METHODS AND RESOURCES

# DAJIN enables multiplex genotyping to simultaneously validate intended and unintended target genome editing outcomes

**Akihiro Kuno**[1,2]☯*, **Yoshihisa Ikeda**[3,4]☯, **Shinya Ayabe**[5]☯, **Kanako Kato**[4], **Kotaro Sakamoto**[2,6], **Sayaka R. Suzuki**[2,7], **Kento Morimoto**[8], **Arata Wakimoto**[1,2], **Natsuki Mikami**[2], **Miyuki Ishida**[4], **Natsumi Iki**[4], **Yuko Hamada**[4], **Megumi Takemura**[1,4], **Yoko Daitoku**[4], **Yoko Tanimoto**[4], **Tra Thi Huong Dinh**[4], **Kazuya Murata**[2,4], **Michito Hamada**[1,4], **Masafumi Muratani**[9], **Atsushi Yoshiki**[5], **Fumihiro Sugiyama**[4], **Satoru Takahashi**[1,4], **Seiya Mizuno**[4]*

**1** Department of Anatomy and Embryology, Faculty of Medicine, University of Tsukuba, Tsukuba, Japan, **2** Ph.D Program in Human Biology, School of Integrative and Global Majors, University of Tsukuba, Tsukuba, Japan, **3** Doctoral Program in Biomedical Sciences, Graduate School of Comprehensive Human Sciences, University of Tsukuba, Tsukuba, Japan, **4** Laboratory Animal Resource Center, Transborder Medical Research Center, Faculty of Medicine, University of Tsukuba, Tsukuba, Japan, **5** Experimental Animal Division, RIKEN BioResource Research Center, Tsukuba, Japan, **6** Department of Computer Science, University of Tsukuba, Tsukuba, Japan, **7** Bioinformatics Laboratory, Faculty of Medicine, University of Tsukuba, Tsukuba, Japan, **8** Doctoral Program in Medical Sciences, Graduate School of Comprehensive Human Sciences, University of Tsukuba, Tsukuba, Japan, **9** Department of Genome Biology, Faculty of Medicine, University of Tsukuba, Tsukuba, Japan

☯ These authors contributed equally to this work.
* akuno@md.tsukuba.ac.jp (AK); konezumi@md.tsukuba.ac.jp (SM)

**Data Availability Statement:** All underlying data can be found in the Supporting Information files deposited at the OSF repository (https://osf.io/w7ade/). DAJIN is accessible at https://github.com/

## Abstract

Genome editing can introduce designed mutations into a target genomic site. Recent research has revealed that it can also induce various unintended events such as structural variations, small indels, and substitutions at, and in some cases, away from the target site. These rearrangements may result in confounding phenotypes in biomedical research samples and cause a concern in clinical or agricultural applications. However, current genotyping methods do not allow a comprehensive analysis of diverse mutations for phasing and mosaic variant detection. Here, we developed a genotyping method with an on-target site analysis software named Determine Allele mutations and Judge Intended genotype by Nanopore sequencer (DAJIN) that can automatically identify and classify both intended and unintended diverse mutations, including point mutations, deletions, inversions, and *cis* double knock-in at single-nucleotide resolution. Our approach with DAJIN can handle approximately 100 samples under different editing conditions in a single run. With its high versatility, scalability, and convenience, DAJIN-assisted multiplex genotyping may become a new standard for validating genome editing outcomes.

akikuno/DAJIN under the MIT Licence. The version of DAJIN used in this study to reproduce the analyses can be found at https://github.com/akikuno/DAJIN/tree/manuscript-version. All sequencing data are available in the DDBJ DRA under accession number DRA011971 (https://ddbj.nig.ac.jp/resource/sra-submission/DRA011971).

**Funding:** Grant number 19H03142 to S.M. and A. K. from the Ministry of Education, Culture, Sports, Science, and Technology. Grant number 20ae0201011h0003 to S.M. and S.T. from the Japan Agency for Medical Research and Development. Grant number JPMJPF2017 to S.T. and A.Y. from the Japan Science and Technology Agency. The funders had no role in study design, data collection and analysis, decision to publish, or preparation of the manuscript.

**Competing interests:** The authors have declared that no competing interests exist.

**Abbreviations:** DAJIN, Determine Allele mutations and Judge Intended genotype by Nanopore sequencer; DNN, deep neural network; FC, fully connected; gRNA, guide RNA; HDBSCAN, hierarchical density-based spatial clustering of applications with noise; KI, knock-in; KO, knockout; LAR, large rearrangement; LOF, local outlier factor; MIDS, Match, Insertion, Deletion, and Substitution; PAM, protospacer adjacent motif; PCA, principal component analysis; PM, point mutation; short-read NGS, short-read next-generation sequencing; SNV, single-nucleotide variant; ssODN, single-strand oligodeoxynucleotide; SV, structural variation; UMAP, Uniform Manifold Approximation and Projection; UMI, unique molecular identifier.

## Introduction

The development of new technologies such as CRISPR-Cas has facilitated genome editing of any species or cell type. Nucleases such as Cas9 and FokI and deaminase fused with Cas9 have been used to introduce DNA double-strand breaks and perform base editing, respectively [1–3]. However, as double-strand break repair pathways are regulated by host cells [4], verifying the result and selecting desired mutated alleles for precise genome editing are essential. Multiple alleles exist in a population of cells or individual animals that have undergone genome editing. In most cases, animals born following editing events at early embryonic stages are mosaic [5]. Heterogeneous cell populations can be obtained by genome editing of cultured cells or delivering genome editing tools to somatic cells [6,7].

Cell populations with incorrectly edited alleles need to be detected and excluded to ensure precise genome editing [8]. Unintended alleles with similar genetic impact may be tolerated only in a specific purpose of genome editing, for instance, generation of null alleles through the deletion of critical exon(s) by using multiple guide RNAs (gRNAs), resulting in multiple patterns in the total deleted length [9]. Recent studies have found that genome editing can induce various on-target events such as inversions, deletions, and endogenous and exogenous DNA insertions as well as indels and substitutions at, and in some cases, away from the target site [10–13]. Furthermore, there is a possibility of gene conversion between homologous regions following genomic DNA cleavage [14–16].

The assessment of on-target editing outcomes and the selection of correct, precisely edited alleles lead to efficient production and breeding of founder animals and their offspring as well as efficient in vivo and ex vivo engineering. Demultiplexing of highly homologous mutated alleles is required to separate the signals of each allele from genetically engineered samples. However, the subcloning of amplified products is laborious, and short-range assessments with targeted PCR amplification and tracking of indels by decomposition analysis of Sanger sequencing data are likely to miss long-range mutation events, which may result in pathogenic phenotypes through unintended changes in gene expression [17,18]. Moreover, short-range PCR analysis followed by illumina-based short-read next-generation sequencing (short-read NGS) cannot identify multiple intended or unwanted mutations in cis or in trans [19,20]. Long-read sequencing technologies enable a comprehensive analysis of the region of interest by providing longer sequence reads compared to the traditional strategy and make it possible to identify unexpected genome editing outcomes, including complex structural variations (SVs) [10,13]. Although targeted long-read sequencing allows the detection of complex on-target mutations over several kilobases [13,21], this method has instrumental limitations such as error rates and lack of tools for phasing and mosaic variant detection to validate multiple and diverse allelic variants to a single-base level [22]. Thus, more accessible and high-throughput methods for routine assessment of genome editing outcomes are essential to detect the unpredictable editing events.

Herein, we describe a novel method for analysing genome editing outcomes, in which long-chain PCR products with barcodes obtained using 2-step long-range PCR were used as samples, and allele validation was performed using our original software named Determine Allele mutations and Judge Intended genotype by Nanopore sequencer (DAJIN) that enables the comprehensive analysis of long reads generated using the nanopore long-read sequencing technology. DAJIN, a machine learning–based model, identifies and quantifies allele numbers and their mutation patterns and reports consensus sequences to visualise mutations in alleles at single-nucleotide resolutions. Moreover, it allows multiple sample processing, and approximately 100 samples can be processed within a day. Because of these features, our strategy with DAJIN can validate the quality of genome-edited samples to select animals or clones with

intended results efficiently and as such has the potential to contribute to more precise genome editing.

## Methods

### Animals

ICR and C57BL/6J mice were purchased from Charles River Laboratories Japan (Yokohama, Japan). C57BL/6J-*Tyr*$^{em2Utr}$ mice were provided by RIKEN BRC (#RBRC06459). Mice were kept in plastic cages under specific pathogen-free conditions in a room maintained at 23.5 ± 2.5˚C and 52.5 ± 12.5% relative humidity under a 14-h light:10-h dark cycle. Mice had free access to commercial chow (MF diet; Oriental Yeast, Tokyo, Japan) and filtered water. All animal experiments were performed humanely with the approval from the Institutional Animal Experiment Committee of the University of Tsukuba following the Regulations for Animal Experiments of the University of Tsukuba and Fundamental Guidelines for Proper Conduct of Animal Experiments and Related Activities in Academic Research Institutions under the jurisdiction of the Ministry of Education, Culture, Sports, Science, and Technology of Japan. The IACUC approval number for this animal experiment was UT_19–003. The euthanasia was performed by cervical dislocation by a skilled person in adult mice and by decapitation with sufficiently keen dissection scissors in newborn mice.

### Genome editing in mouse zygotes

Mice with point mutations (PMs) and 2-cut knockout (KO) were generated using the electroporation method [23]. The gRNA target sequences to induce each mutation are listed in S1 Table. The gRNAs were synthesised and purified using a GeneArt Precision gRNA Synthesis Kit (Thermo Fisher Scientific, Waltham, MA, USA) and dissolved in Opti-MEM (Thermo Fisher Scientific). In addition, we designed 3 single-strand oligodeoxynucleotides (ssODNs) donors for inducing PMs in *Tyr* (S1 Table). These ssODN donors were ordered as Ultramer DNA oligos from Integrated DNA Technologies (Coralville, IA, USA) and dissolved in Opti-MEM. The mixtures of gRNA (5 ng/μL) and ssODNs (100 ng/μL) or mixtures of 2 gRNAs (25 ng/μL each) were used to generate point mutant mice or 2-cut KO mice, respectively. GeneArt Platinum Cas9 Nuclease (100 ng/μL; Thermo Fisher Scientific) was added to these mixtures. Pregnant mare serum gonadotropin (5 units) and human chorionic gonadotropin (5 units) were intraperitoneally injected into female C57BL/6J mice (Charles River Laboratories) with a 48-h interval. Next, unfertilised oocytes were collected from their oviducts. Then, according to standard protocols, we performed in vitro fertilisation with these oocytes and sperm from male C57BL/6J mice (Charles River Laboratories). After 5 h, the abovementioned gRNA/ssODN/Cas9 or 2 gRNAs/Cas9 mixtures were electroplated into the mouse zygotes using a NEPA 21 electroplater (NEPAGNENE; Chiba, Japan), under previously reported conditions [24]. The electroporated embryos that developed into the 2-cell stage were transferred to oviducts of pseudopregnant ICR female mice. The floxed mice were generated using the microinjection method [25]. Each gRNA target sequence (S1 Table) was inserted into the entry site of pX330-mC carrying both the gRNA and Cas9 expression units. These pX330-mC plasmid DNAs and donor DNA plasmid were isolated using FastGene Plasmid Mini kit (Nippon Genetics, Tokyo, Japan) and filtered using MILLEX-GV 0.22 μm filter unit (Merck Millipore, Darmstadt, Germany) for microinjection. Next, C57BL/6J female mice superovulated using the method described above were naturally mated with male C57BL/6J mice, and zygotes were collected from the oviducts of the mated female mice. For each gene, a mixture of 2 pX330-mC (circular, 5 ng/μL each) and a donor (circular, 10 ng/μL) was microinjected into the zygote.

The zygotes that survived were then transferred into the oviducts of pseudopregnant ICR female mice. When the newborns were around 3 weeks of age (S2 Table), the tail was sampled to obtain genomic DNA.

## Library preparation and nanopore sequencing

We used PI-200 (Kurabo Industries, Osaka, Japan), according to the manufacturer's protocol, for the extraction and purification of genomic DNA obtained from the tail of mice. The purified genomic DNA was amplified using PCR using KOD multi & Epi (Toyobo, Osaka, Japan) and target amplicon primers (S3 Table). In the target amplicon primer, the universal sequence (22 mer) is located on the 5′ side, and the sequence for target gene amplification is on the 3′ side. Five-fold dilutions of the PCR products were used as templates for nested PCR performed using KOD multi & Epi and barcode attachment primers (S4 Table). The 5′ side of the barcode attachment primer has a barcode sequence (24 mer), and the 3′ sequence is annealed to the universal sequence of the target amplicon primer (Fig 1A). The barcoded PCR products were mixed in equal amounts and then purified using FastGene Gel/PCR Extraction Kit (Nippon Genetics, Germany). The volume of the mixed and purified PCR products was adjusted to 20 to 30 ng/μL. The library was prepared using Ligation Sequencing 1D kit (SQK-LSK108_109; ONT, Oxford, UK) and NEBNext End repair/dA-tailing Module NEB Blunt/TA Ligase Master Mix (New England Biolabs, Ipswich, MA, USA) according to the manufacturer's instructions. The prepared library was loaded onto a primed R9.4 Spot-On Flow cell (FLO-MIN106; ONT, Oxford, UK). The 24-h or 36-h run time calling sequencing protocol was selected in the Min-KNOW GUI (version 4.0.20), and base calling was allowed to complete after the sequencing run was completed. After base calling, we demultiplexed the barcoding libraries using qcat (version 1.1.0) with default parameter settings. Total nanopore sequencing reads per sample are listed in S5 Table.

## Conventional genotyping analysis

To evaluate the validity of DAJIN's genotyping results, we used conventional genotyping methods, including short-amplicon PCR, PCR-RFLP, and Sanger sequencing. For the genotyping of the 2-cut KO and PM lines, genomic PCR was performed using AmpliTaq Gold 360 DNA Polymerase (Thermo Fisher Scientific) and the relevant primers (S6 Table). Agarose gel electrophoresis was performed to confirm the size of the PCR products. In the flox knock-in (KI) design, genomic PCR was performed using KOD FX (Toyobo) and the relevant primers (S6 Table). The PCR products were digested with restriction enzymes *Asc*I (New England Biolabs) and *Eco*RV (New England Biolabs) for 2 h to check LoxP insertion on the left and right side, respectively. Agarose gel electrophoresis was performed to confirm the size of the PCR fragments. PCR products with mutant sequences were identified using Sanger sequencing using the BigDye Terminator v3.1 Cycle Sequencing Kit (Thermo Fisher Scientific).

## Targeted next-generation sequencing

Genomic PCR was performed using AmpliTaq Gold 360 DNA Polymerase (Thermo Fisher Scientific) and the relevant primers whose barcode sequences were added to the 5′ end (S7 Table). The PCR amplicons in 226 bp (*Tyr*.c140 G>C) and 203 bp (*Tyr* c.316 G>C, *Tyr* c.308 G>C) lengths were purified using FastGene Gel/PCR Extraction Kit (Nippon Genetics, Düren, Germany). Paired-end sequencing (2 × 151 bases) with these purified amplicons was performed using MiSeq (Illumina, San Diego, CA, USA) at Tsukuba i-Laboratory LLP (Tsukuba, Ibaraki, Japan). Paired-end reads were mapped against chromosome 7 of mouse genome

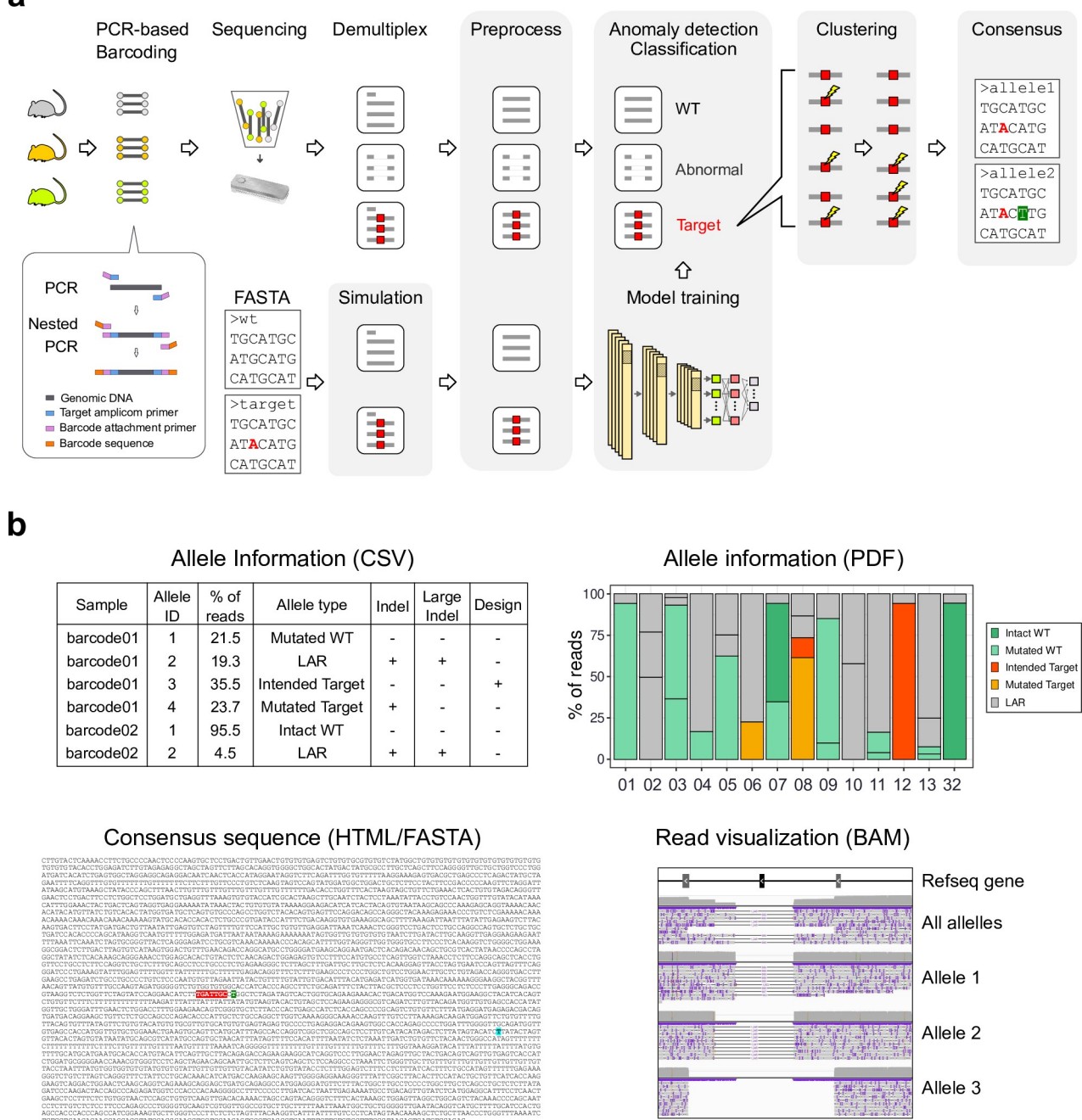

**Fig 1. Overview of the methods.** (a) The schematic of DAJIN's workflow. DAJIN automates the procedures highlighted in grey. Red-coloured nucleotides represent intended PM. A green-highlighted nucleotide represents unintended substitution. Illustrations were modified from the Togo picture gallery, licenced under CC-BY 4.0 Togo picture gallery by the DBCLS, Japan. (b) The outputs of DAJIN. The file formats are described in parentheses. See S1 Data for raw data from https://osf.io/w7ade/. DAJIN, Determine Allele mutations and Judge Intended genotype by Nanopore sequencer; DBCLS, Database Center for Life Science; LAR, large rearrangement; PM, point mutation; WT, wild type.

assembly mm10 using STAR (version2.7.0a) with default settings [26]. Mapped reads were visualised using IGV (version 2.9.4) [27]. The samples carrying the intended PM at frequencies >10% were considered as positive.

## Nanopore read simulation

To prepare training data for deep neural network (DNN) models, we generated simulation reads of the possible alleles using NanoSim (version 2.5.0) [28]. We trained NanoSim to obtain an error profile using nanopore sequencing reads from a wild-type (WT) control. Next, we applied the error profile to generate 10,000 simulation reads per each possible allele that could be caused by genome editing (S1 Fig). In the PM design, we generated simulation reads with a deletion or random nucleotide insertion of the gRNA length at the Cas-cutting site.

## Preprocessing

We performed preprocessing to exclude reads without target loci and to perform Match, Insertion, Deletion, and Substitution (MIDS) conversion. First, the genome-edited sequence was aligned to the user-provided WT sequence using minimap2 (version 2.17) [29] with the "—cs = long" option, and the position of the target mutant base was detected according to the CS-tag in the SAM file. Simulated and nanopore sequencing reads were then aligned using minimap2 to the WT sequence. Reads with lengths more than 1.1 times longer than the maximum length among possible alleles were excluded. For the remaining reads, we detected the start and end positions of each read relative to the WT sequence based on CIGAR information and extracted the reads containing the mutant region of interest (S4A Fig).

The extracted reads were subjected to MIDS conversion (S4B Fig). The matched, inserted, deleted, and substituted bases compared to the control sequence were converted to M (Match), I (Insertion), D (Deletion), and S (Substitution), respectively. Next, the read lengths were trimmed or padded with "=" to equalise their sequence length. Then, one-hot encoding was performed on the MIDS sequence.

## Deep learning model

We constructed a DNN model to classify alleles. The structure of the deep learning model is shown in S5 Fig. The architecture comprises 3 layers of convolutional and max-pooling layers and a fully connected (FC) layer, and a softmax function to predict the allele types. The batch size was 32. The maximum number of training epochs was 200, and the training was stopped when validation loss was not improved during 20 epochs. To detect reads with large rearrangements, we extracted the outputs from the FC layer. Then, we trained the local outlier factor (LOF) [30] using the output of the simulated reads. Subsequently, the output of the nanopore sequence reads was placed in the LOF; it annotated unexpected mutation reads as "large rearrangements (LARs)," which we define in this manuscript the name of genomic rearrangements more than around 50 bp in length. We assessed the accuracy of the classification using simulation reads, and it was able to accurately classify alleles in all genome editing designs conducted in this study (S8 Table).

## Allele clustering

In order to distinguish each allele precisely, DAJIN conducts compressed MIDS conversion and clustering. To generate fixed-length sequences, we performed compressed MIDS conversion, which replaces successive insertions with a character corresponding to the number of insertions and then substitutes the insertion (S6 Fig). A character is assigned to the number from 1 to 9 or a letter from a to z. If the number of consecutive insertions is in the range 1 to 9, the character is the corresponding number. If the number of consecutive insertions is in the range 10 to 35, the character is "a" (= 10) to "y" (= 35). If the number is greater than 35, the character is "z" (>35).

To mitigate nanopore sequencing errors, the MIDS's relative frequencies of the sample were subtracted from the control reads. We call the subtracted MIDS relative frequencies "MIDS score" (S7 Fig). The MIDS score was reduced into 5 dimensions using principal component analysis (PCA). Then, hierarchical density-based spatial clustering of applications with noise (HDBSCAN) [31] was performed for the allele clustering. For parameters, we set "min_-samples," which specifies the minimum size of each cluster formed, as "1" to maximise valid reads. Furthermore, we tuned "min_cluster_size," which defines the minimum number of samples in each cluster. We set the value as 50 equal intervals between 10% and 40% of the total number of reads and then selected the "min_cluster_size," which outputs the mode of cluster numbers.

### Filter minor alleles

To improve the interpretability, DAJIN has a default setup to remove minor alleles. Minor alleles were defined as those in which the number of reads was 1% or less of the total number of reads of a sample. DAJIN was able to report all allele information using the "filter = off" option.

### Consensus sequence

The consensus sequence for each allele was output as a FASTA file and an HTML file. In the HTML file, the mutated nucleotides are coloured. To generate the consensus sequence, we compared FASTA alleles and compressed MIDS sequences.

### Generation and visualisation of BAM files

DAJIN generates BAM files to visualise the DAJIN-reported alleles in a genome browser. First, DAJIN uses minimap2 to map the nanopore sequence reads to the WT sequences described in the user-inputted FASTA file, then samtools (version 1.10) [32] generates sorted BAM files. Next, the target genome coordinates and chromosome length are obtained from the UCSC Table Browser [33] according to the user-inputted FASTA file and genome assembly ID. Then, DAJIN replaces the chromosome number and chromosome length in SN and LN headers of BAM files.

### Single-nucleotide variant (SNV) and structural variation (SV) callers

We installed Medaka (version 1.2.1) [34], Clair (version 2.1.1) [35], NanoCaller (version 1.0.0) [36], NanoSV (version 1.2.4) [37], NGMLR (version 0.2.7) [38], Sniffles (version 1.0.12) [38] via Bioconda [39]. For Sniffles, the parameters of "—cluster" and "-n -1" were provided to phase SVs in all sequencing reads; otherwise, the default parameters were chosen.

## Results

### Workflow of DAJIN

We designed DAJIN to genotype genome-edited samples by capturing diverse mutations from SNVs to LARs that covers genomic rearrangements more than approximately 50 bp in length. The overall workflow of DAJIN is presented in Fig 1A. DAJIN requires (1) a FASTA file describing possible alleles, which must include the DNA sequence before and after genome editing; (2) FASTQ files from nanopore sequencing, which include a control sample; (3) gRNA sequence including the protospacer adjacent motif (PAM); and (4) a genome assembly ID such as hg38 and mm10. Next, DAJIN generates simulation reads using NanoSim [28] according to the user-inputted FASTA file. The sequence reads are preprocessed and one-hot

encoded. Subsequently, the simulated reads are used to train a DNN model to detect LAR reads and classify allele types. DAJIN defines LAR alleles as a different sequence from the user-inputted FASTA file. Next, DAJIN conducts clustering to estimate the alleles. Finally, it reports the consensus sequence to visualise the mutations in each allele and labels the alleles. The details are described in Methods and S1 and S4–S7 Figs. The outputs of DAJIN are shown in Fig 1B. DAJIN reports allele frequencies in each sample, the consensus sequences, and BAM files for each allele. In this study, DAJIN was evaluated on 9 mouse strains of 3 types of genome editing design: PM, 2-cut KO, and flox KI. The performance evaluations are described in detail below.

## Performance of LAR detection

CRISPR-Cas genome editing has been reported to induce unexpectedly large indels, which might be overlooked by conventional short PCR-based genotyping methods. Conversely, a nanopore long-read sequencer can capture large indels within its amplicon size, which allows the detection of LAR alleles. Thus, we implemented a LAR detection using DNN (see Methods in detail) into DAJIN and evaluated its performance. We simulated nanopore reads with a deletion in the range of 10 bp to 200 bp at the *Cables2* genome locus (S2A Fig). Next, we pre-processed the simulated reads with or without MIDS conversion and conducted Uniform Manifold Approximation and Projection (UMAP) [40] and LOF [30] using outputs from the last FC layer (S2B Fig). The UMAP revealed the cluster of 50 bp deletion with MIDS conversion, which was unclear without MIDS conversion (S2C Fig). We next investigated the accuracy of LAR allele detection. The results showed that DAJIN labelled more than 50 bp deletion as LAR with MIDS conversion, but not without, which indicates that the MIDS conversion improves LAR allele detection (S2D Fig). Since several SV callers using long-read sequencing have been developed, we next compared DAJIN to NanoSV [37] and Sniffles [38]. We prepared 1000 simulated reads containing deletions of 50, 100, and 200 bases for each allele and mixed them to imitate genome-edited samples with the 3 alleles (S3A Fig). Then, we evaluated the samples using DAJIN, NanoSV, and Sniffles. The results showed that DAJIN discriminated each allele according to the deletion sizes; however, NanoSV and Sniffles did not (S3B Fig).

## DAJIN captures point mutation alleles

Next, we evaluated DAJIN's performance using genome-edited mice. We induced *Tyr* c.140G>C PM using CRISPR-Cas9 genome editing in C57BL/6J mouse fertilised eggs and obtained 13 founder mice. Next, we amplified a 2,845-bp DNA sequence at the *Tyr* loci of the founder mice (barcode (BC) 01 to 13) and a WT control mouse (BC32) (Fig 2A). Then, DAJIN was used to analyse the PCR amplicons of the 14 mice (Fig 2B). Because the PM's genome editing design potentially generates WT, PM, and unexpected SV alleles, DAJIN annotated "WT," "PM," and "LAR" allele types. Besides, when DAJIN's consensus sequence perfectly matched the sequences of "WT" and "PM" described in the user-inputted FASTA file, these alleles were labelled as "Intact WT" and "Intended PM," respectively, whereas when there was a mismatch between DAJIN's consensus sequence and FASTA sequence, it was labelled as "Mutated WT" and "Mutated PM."

DAJIN reported the percentages of the predicted allele types and identified 2 mice (BC08 and BC12) having the intended PM (Fig 2B). Visualisation using IGV showed that DAJIN accurately captured the c.140G>C PM in BC08 and BC12 (Fig 2C). Moreover, DAJIN detected an unexpected 2-bp insertion in BC08 at 23 bp downstream from the PM and labelled the allele as "Mutated PM" (Fig 2C). Next, DAJIN's consensus sequence reported the BC12 included in the "Intended PM" allele (Fig 2D). Sanger sequencing of BC12 at the PM locus supported the

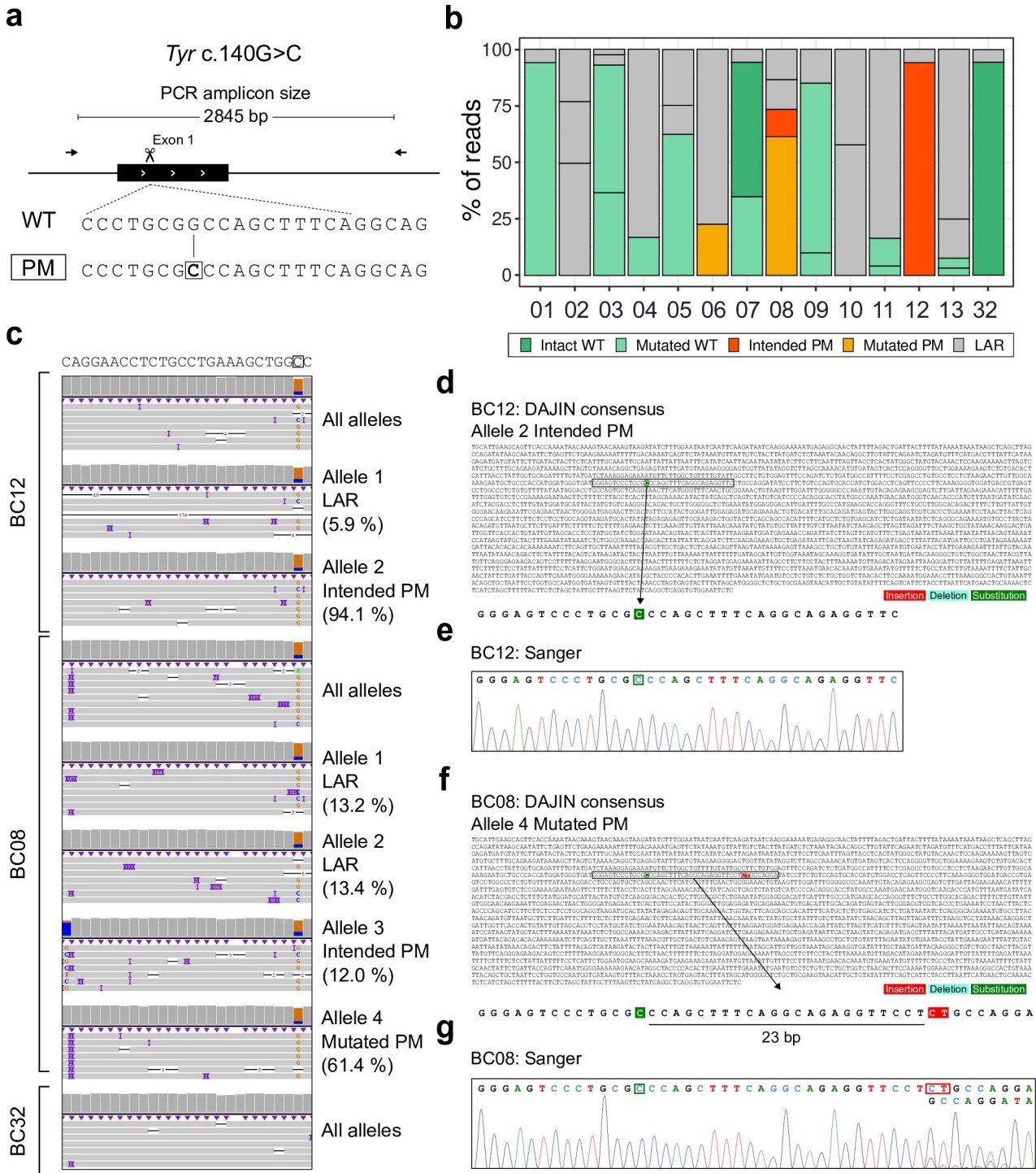

**Fig 2. Application of DAJIN for PM design.** (a) Genome editing design for Tyr c.140G>C PM. The scissors represent a Cas9-cutting site. The arrows represent PCR primers, including the PCR amplicon size. The boxed allele type represents the target allele, and the boxed nucleotide represents a targeted PM. (b) DAJIN's report of the allele percentage. The barcode numbers on the x-axis represent mouse IDs. BC32 is a WT control. The y-axis represents the percentage of DAJIN-reported alleles. The colours of the bars represent DAJIN-reported allele types. The compartments partitioned off by horizontal lines in a bar represent the DAJIN-reported alleles. (c) Visualisation of nanopore sequencing reads at Tyr target locus. BC12 and BC08 contain target alleles. BC32 is a WT control. The "All alleles" track represents all reads of each sample. The "Allele" track represents DAJIN-reported alleles. (d) Comparison between DAJIN's consensus sequence and Sanger sequencing. The sequence represents the consensus sequence of a dominant allele of BC12 and BC08. The colours on the nucleotides represent mutation types, including insertion (red), deletion (sky blue), and substitution (green). The coloured boxes in the Sanger sequence represent mutated nucleotides, including insertion (red) and substitution

(green). See S2 Data for raw data from https://osf.io/w7ade/. DAJIN, Determine Allele mutations and Judge Intended genotype by Nanopore sequencer; LAR, large rearrangement; PM, point mutation; WT, wild type.

target PM's induction (Fig 2E). Furthermore, DAJIN's consensus sequence of "Mutated PM" allele in BC08 included an unexpected 2-bp (CT) insertion as well as the PM (Fig 2F). The same "CT" insertion was validated via Sanger sequencing (Fig 2G). Besides, DAJIN reported the ratio of "CT" inserted alleles and the other alleles was approximately 5:1 (Fig 2C). Sanger sequencing also showed a waveform intensity ratio of 5:1 between the PM with CT insertion allele and that without the insertion allele (Fig 2G). This consistency indicated that DAJIN accurately quantifies the allele frequencies.

To further evaluate DAJIN's performance, we generated 2 more PM mice, *Tyr* c.316G>C and *Tyr* c.308G>C (S8 Fig). Besides, a C57BL/6J-*Tyr*^em2Utr mouse, which carries the *Tyr* c.230G>T PM [5], was added to BC31 as a control for "Mutated WT." For *Tyr* c.316G>C and c.308G>C projects, DAJIN reported that 1 out of 6 mice (BC18) and 8 out of 11 mice (BC21, BC22, BC23, BC24, BC26, BC29, and BC30) had the "Intended PM" (Fig 3A). As with BC12, DAJIN annotated almost all (93.6%) nanopore sequencing reads in BC21 as "Intended PM." Subsequently, DAJIN's consensus sequence in BC21 reported the intended c.308G>C PM, and we detected a single waveform of the PM using Sanger sequence analysis (S9 Fig). In addition, DAJIN correctly identified *Tyr* c.230G>T PM in BC31, which was used as the positive control of "Mutated PM" (S10 Fig).

Next, we genotyped the PM mice using short-read NGS and compared them to DAJIN's results. Short-read NGS reported that a total of 16 mice (BC06, BC08, and BC12 in c.140G>C; BC14, BC15, and BC18 in c.316G>C; BC20, BC21, BC22, BC23, BC24, BC25, BC26, BC27, BC29, and BC30 in c.316G>C) had the intended PM (Fig 3A). Notably, we found that the genotyping of short-read NGS can be misleading when the samples contain LAR alleles. For instance, DAJIN reported BC20 as a mosaic, including 2 LARs, 1 intended PM, and 1 mutated WT, and we confirmed the alleles via Sanger sequencing (Fig 3B and 3C). In contrast, because short-read NGS could not capture LARs, the genotyping result seemed heterozygous (Fig 3C). Next, DAJIN showed BC25 included 2 LAR alleles (approximately 70 bp insertion and large deletion). On the other hand, short-read NGS showed 1 allele, which may be derived from the approximately 70 bp inserted allele (Fig 3D). Furthermore, as with BC20, DAJIN reported BC26 as a mosaic including LAR alleles, while short-read NGS reported it as heterozygous (Fig 3E). Besides, the mapping percentages of BC20, BC25, and BC26 were approximately 97% to 99%, which suggests that the preparing short amplicon via PCR and the limited number of cycles in short-read NGS might be the main reason for the impaired LAR allele detection (S9 Table). These results indicate that although short-read NGS can capture PM with high sensitivity and specificity, excluding LAR alleles by long-read sequencing is essential for accurate genotyping.

To compare DAJIN to existing long-read–based SNV callers, we performed Medaka and Clair [35] on the 16 mice reported as PM positive by short-read NGS. The results showed that both Medaka and Clair were prone to overlook the minor PM alleles (S10 Table), which may be because they were designed to handle diploid genomes, while DAJIN can treat multiallelic mutations. Although NanoCaller [36] showed the best performance to detect PM alleles, it was not able to report LAR with PM alleles in BC25 (S10 Table).

We next evaluated whether DAJIN correctly captured LAR alleles. We conducted short and long PCRs for detecting small and large indel mutations (S11A Fig). The PCRs revealed 17 samples with aberrant PCR bands, which were consistent with the samples with LAR alleles reported by DAJIN (S11B and S11C Fig). We further analysed BC02 and BC10, the alleles

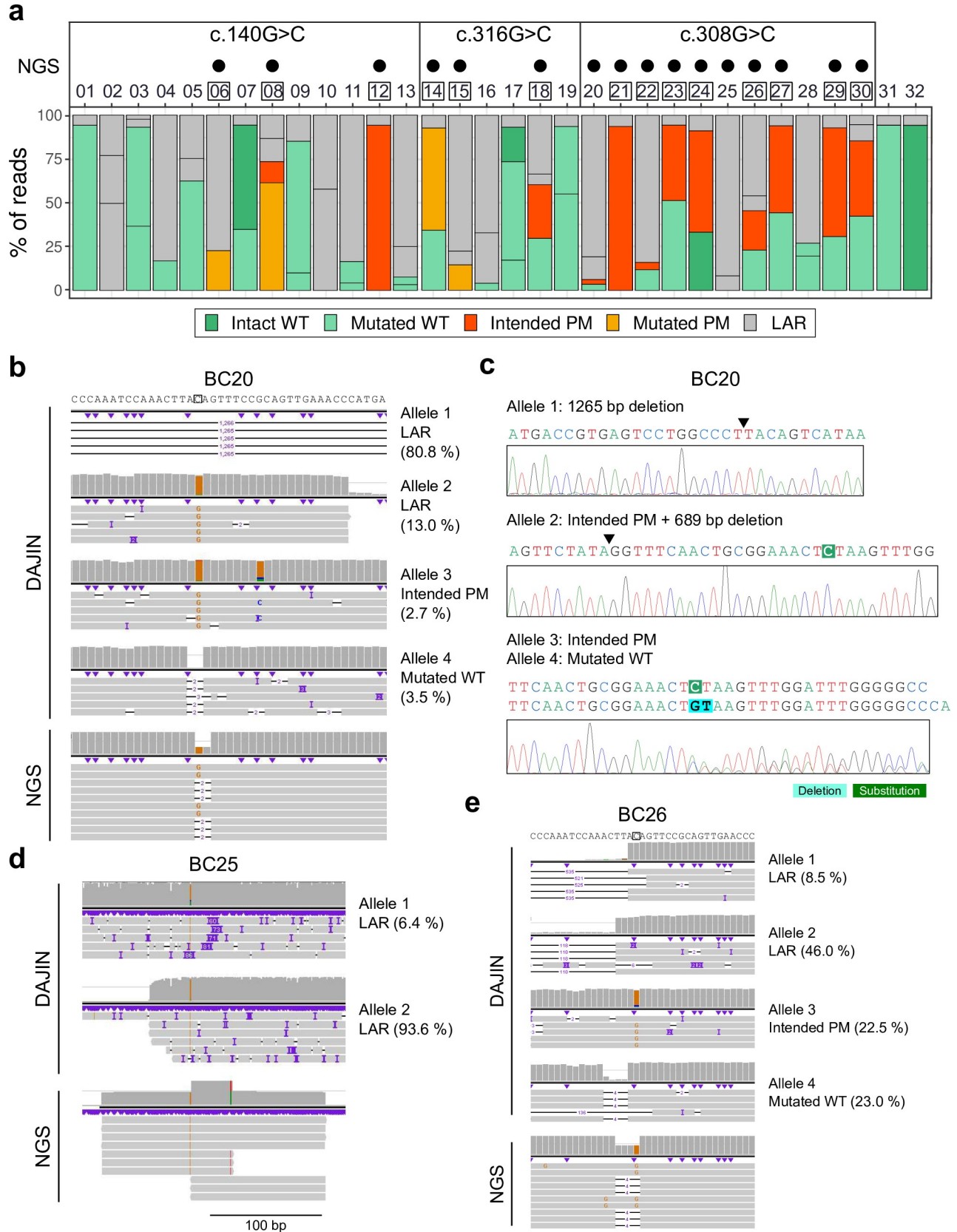

**Fig 3. Comparison to short-read NGS for PM design.** (a) The genotyping results obtained using DAJIN and short-read NGS. The numbers on the x-axis represent barcode IDs. The genome editing of BC01–BC13, BC14–BC19, and BC20–BC30 aims to induce c.140G>C, c.316G>C, and c.308G>C, respectively. The black dots represent PM-positive samples detected using short-read NGS. The boxed numbers represent PM-positive samples detected using DAJIN. The barplot of c.140G>C samples is the same plot as shown in Fig 2B. The BC31 and BC32 are albino (c.230G>T) and WT mice, respectively. The y-axis represents the percentage of DAJIN-reported alleles. The colours of the bar represent DAJIN-reported allele types. The horizontal lines in a bar represent the DAJIN-reported alleles. (b) Comparison to short-read NGS in BC20. (c) Sanger verification of DAJIN-detected alleles. The arrowheads represent the junction sites of DNA. (d) Comparison to short-read NGS in BC25. (e) Comparison to short-read NGS in BC26. See S3 Data for raw data from https://osf.io/w7ade/. DAJIN, Determine Allele mutations and Judge Intended genotype by Nanopore sequencer; LAR, large rearrangement; PM, point mutation; short-read NGS, short-read next-generation sequencing; WT, wild type.

reported as "LAR" by DAJIN. Visualisation of the reads revealed that BC02 and BC10 had approximately 50 bp and 40 bp insertions, respectively (S12A Fig). We conducted PCR and validated that BC02 and BC10 had 50 bp and 40 bp larger bands, respectively, than that in WT control (S12B and S12C Fig). This result indicated that DAJIN was able to correctly annotate alleles with 40 to 50 bp insertion as "LAR" alleles. Taken together, these results indicated that DAJIN's genotyping outperforms the conventional methods in accurately identifying the PMs owing to LAR allele detection ability.

## DAJIN identifies knockout alleles

We next applied DAJIN to the KO design. We designed to remove exon 6 of *Prdm14* by 2-cut strategy with CRISPR-Cas9 system [25] (Fig 4A). The predicted deletion size was 1,043 bp length and may yield an inverted allele as a by-product. Thus, DAJIN annotated "WT," "Deletion (Del)," "Inversion," and "LAR" alleles. We generated 10 *Prdm14* deletion founder mice (BC16 to BC25) and analysed them using DAJIN with a WT mouse as a control (BC26); of the 10 mice, 5 (BC16, BC18, BC20, BC23, and BC24) contained "Mutated Del" allele (Fig 4B). Next, we evaluated BC18 and BC23 as DAJIN predicted that they contained the "Mutated Del" allele. Visualisation showed that DAJIN discriminated "LAR" alleles with 100 to 200 bp larger deletion than the intended deletion (Fig 4C). Furthermore, DAJIN's consensus of "Mutated Del" alleles showed that BC18 had a 1-bp deletion and that BC23 included the 7-bp insertion and 1-bp substitution at the joint site, respectively. The same mutations were validated using Sanger sequencing (Fig 4D and 4E).

We evaluated the phenotypes of BC18 and BC23 mice. The deletion of *Prdm14* inhibits primordial germ cell differentiation and causes the complete depletion of germ cells in adult female and male mice [41]. We performed immunostaining of testis sections for PLZF1 (spermatogonia marker) and Vimentin (Sertoli cell marker). Spermatogonia were not detected in BC18 and BC23 (Fig 4F).

To confirm whether DAJIN is applicable to the CRISPR-Cas12a system [42], we generated *Prdm14* KO mice using Cas12a (S13A Fig). In all, 15 founder mice were obtained, and DAJIN analysis revealed that 4 of them (BC10, BC11, BC12, and BC13) had "Mutated Del" (S13B Fig). We validated the mice carrying the deletion allele using conventional PCR and electrophoresis analysis. The electrophoresis analysis of the Cas9 group showed that 5 mice (BC16, BC18, BC20, BC23, and BC24) had the deletion alleles, similar to DAJIN's report (S13C and S13D Fig).

To evaluate the versatility of DAJIN at other genomic loci and different cleavage widths, we established KO mice for the *Ddx4* gene using the 2-cut strategy with Cas9 and Cas12a system. *Ddx4* KO was designed to cleave 3,377 bp, including exons 11 to 15. We obtained 21 founder mice and analysed the 5,221-bp PCR amplicon containing the target region (S14A Fig). DAJIN reported that 1 mouse (BC27) subjected to Cas12a-based genome editing and 4 mice (BC36, BC39, BC44, and BC46) subjected to Cas9-based genome editing carried the "Mutated Del" allele (S14B Fig). The presence of the deletion alleles was confirmed via electrophoresis of

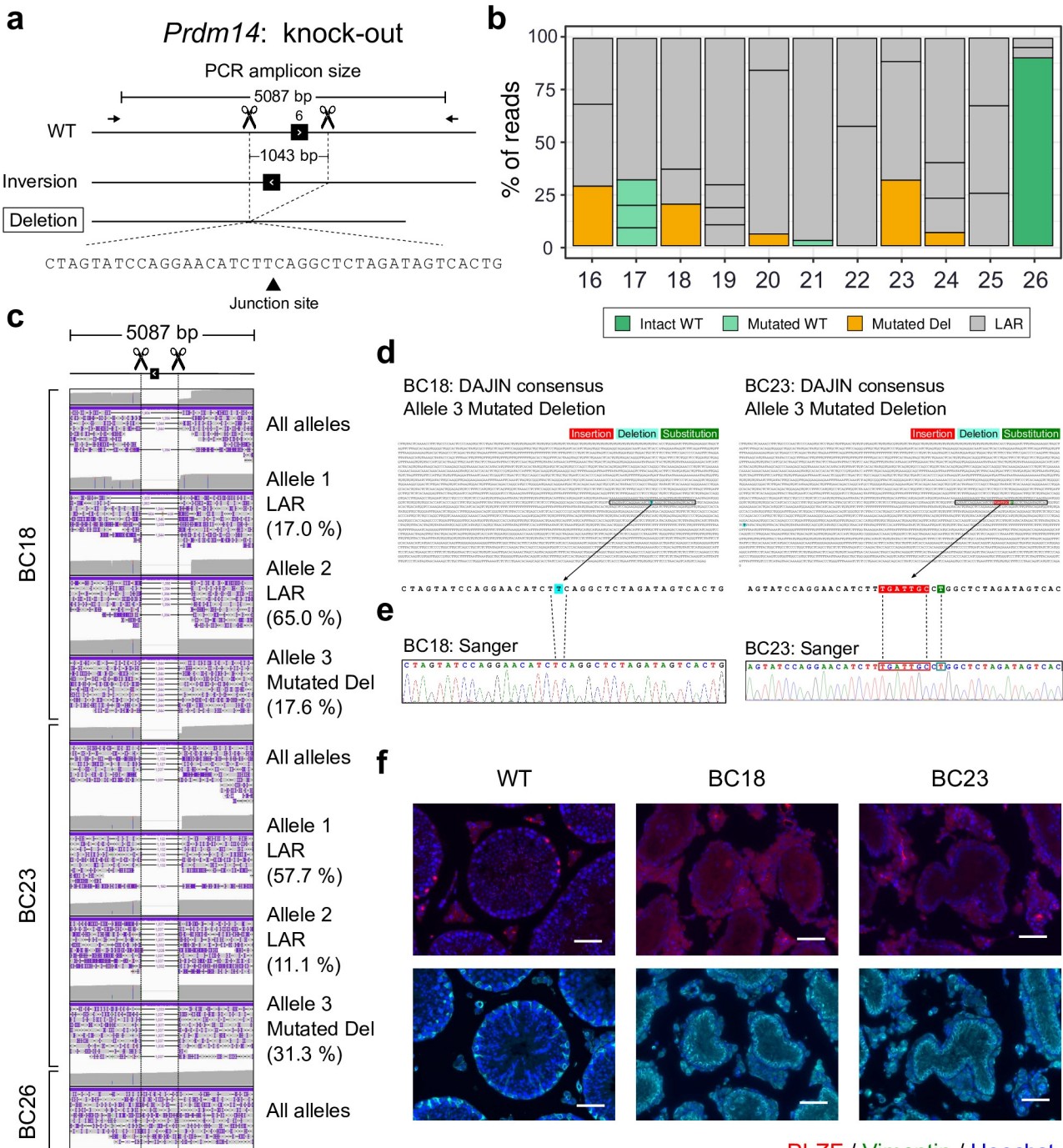

**Fig 4. Application of DAJIN for KO design.** (a) Genome editing design for Prdm14 KO. The scissors and dotted lines represent Cas9-cutting sites. The arrows represent PCR primers, including the size of the PCR amplicon. The boxed allele type represents the target allele. The inversion allele represents a possible by-product. The triangle on the nucleotides represents a junction site of 2 DNA fragments. (b) DAJIN's report of the allele percentage. The barcode numbers on the x-axis represent mouse IDs. BC26 is a WT control. The y-axis represents the percentage of DAJIN-reported alleles. The colours of the bar represent DAJIN-reported allele types. The compartments partitioned off by horizontal lines in a bar represent the DAJIN-reported alleles. (c) Visualisation of nanopore sequence reads at Prdm14 target locus. BC18 and BC23 contain target alleles. BC26 is a WT control. The "All alleles" track represents all reads of each sample. The "Allele" track represents DAJIN-reported alleles. (d) DAJIN's consensus sequences of the target allele. The top sequence represents the consensus sequence of target alleles of BC18 Allele 3 and BC23 Allele 3. The bottom sequences enlarge the boxed sequence of the consensus sequence. The colours on the nucleotides represent mutation types, including insertion (red), deletion (sky blue), and substitution (green). (e) Validation by Sanger sequencing. The dotted lines represent corresponding nucleotides between Sanger and DAJIN's consensus sequences. (f) PLZF (red), Vimentin (green), and Hoechst (blue) staining of the testis section of WT (left), BC18

(middle), and BC23 (right). Upper panels show costaining of PLZF (red) and Hoechst (blue). Lower panels show costaining of Vimentin (green) and Hoechst (blue). PLZF and Vimentin are markers of undifferentiated spermatogonia and Sertoli cells, respectively, along the seminiferous tubules' basal lamina. Scale bar: 100 µm. See S4 Data for raw data from https://osf.io/w7ade/. DAJIN, Determine Allele mutations and Judge Intended genotype by Nanopore sequencer; KO, knockout; LAR, large rearrangement; WT, wild type.

the PCR products (S14C and S14D Fig). Furthermore, we performed detailed genome analysis of the *Stx2* KO mice, which was generated past research using the 2-cut strategy to cleave 727 bp, including exon 5 of *Stx2* [43]. DAJIN reported that 13 of the 29 founder mice (BC01, BC03, BC04, BC05, BC07, BC09, BC14, BC15, BC20, BC21, BC22, BC23, and BC24) had the "Mutated Del" allele (S15A and S15B Fig), and DAJIN's results were consistent with the PCR-based genotyping (S15C and S15D Fig). In the *Stx2* analysis, DAJIN detected the "Inversion" allele in 3 mice (BC08, BC16, and BC17). To verify the inversion alleles, we performed PCR for amplifying the genome region containing the inversion junction sites (S15E Fig). The inversion band was found in all 3 mice (S15F Fig). Besides, the 1-bp (A) insertion at the inversion junction site was found in DAJIN's consensus sequence of BC17. This insertion was also confirmed via Sanger sequencing (S15G Fig). These results indicated that DAJIN could accurately identify SNVs in inversion alleles. Next, DAJIN reported 3 LARs in BC25 (S15B Fig). The PCR electrophoresis validated the 3 alleles (S16A and S16B Fig). Moreover, DAJIN's consensus sequence of BC25 reported that Allele 1, 2, and 3 included 986 bp deletion, 2,477 bp deletion plus 1 bp substitution, and 1,345 bp deletion, respectively (S1 File). Then, we performed Sanger sequencing, and the results perfectly matched with that of DAJIN's report at a single-nucleotide resolution (S16C Fig). Lastly, to compare DAJIN's LAR detection ability to the previous SV callers, we tested NanoSV and Sniffles for the alleles of BC25. Although NanoSV and Sniffles annotated LARs, they were not able to discriminate the 3 alleles correctly (S17 Fig). We also performed NanoSV and Sniffles on BC18 and BC23 in order to test whether they can detect KO alleles. They captured the one cutting site (chr1:13118480) but could not differentiate the intended deletion and LAR alleles with 100 to 200 bp larger deletion than the intended deletion (S2 File). Taken together, these results demonstrated DAJIN's ability to accurately genotype the KO design. DAJIN's genotyping for KO alleles was perfectly matched with Sanger sequencing. Furthermore, it correctly identifies multiallelic LARs, which current tools could not.

## DAJIN identifies flox knock-in alleles

Cre-LoxP–based conditional KO experiments are mostly performed to analyse gene function under specific conditions. Genome editing for generating floxed alleles requires *cis* KI at 2 loci simultaneously, which lowers the generation efficiency. Moreover, genotyping of the *cis* KI is difficult and error prone owing to the need to identify *cis* mutations at several kilobases of the DNA region. Besides, the generation of floxed alleles using ssODNs as the donor of the KI sequence occasionally leads to the introduction of unintended mutations in a critical LoxP sequence because of the error in the synthesis process and its secondary structure [44]. Because of these difficulties, no standard genotyping method is currently available to comprehensively and accurately evaluate flox mutations induced by genome editing in 1 step. Therefore, we evaluated whether DAJIN can correctly genotype floxed alleles at single-nucleotide resolution.

We performed validation experiments using plasmid vectors with completely defined sequences. We generated 6 types of plasmids with LoxP sequences: (1) "Intended flox"; (2) 1-bp insertion in left LoxP; (3) 1-bp deletion in left LoxP; (4) 1-bp substitution in left LoxP; (5) 1-bp substitution in right LoxP; and (6) 1-bp substitution in both LoxPs (Fig 5A). We mixed the WT genomic DNA with each plasmid to imitate the heterozygous genotype. The mixed

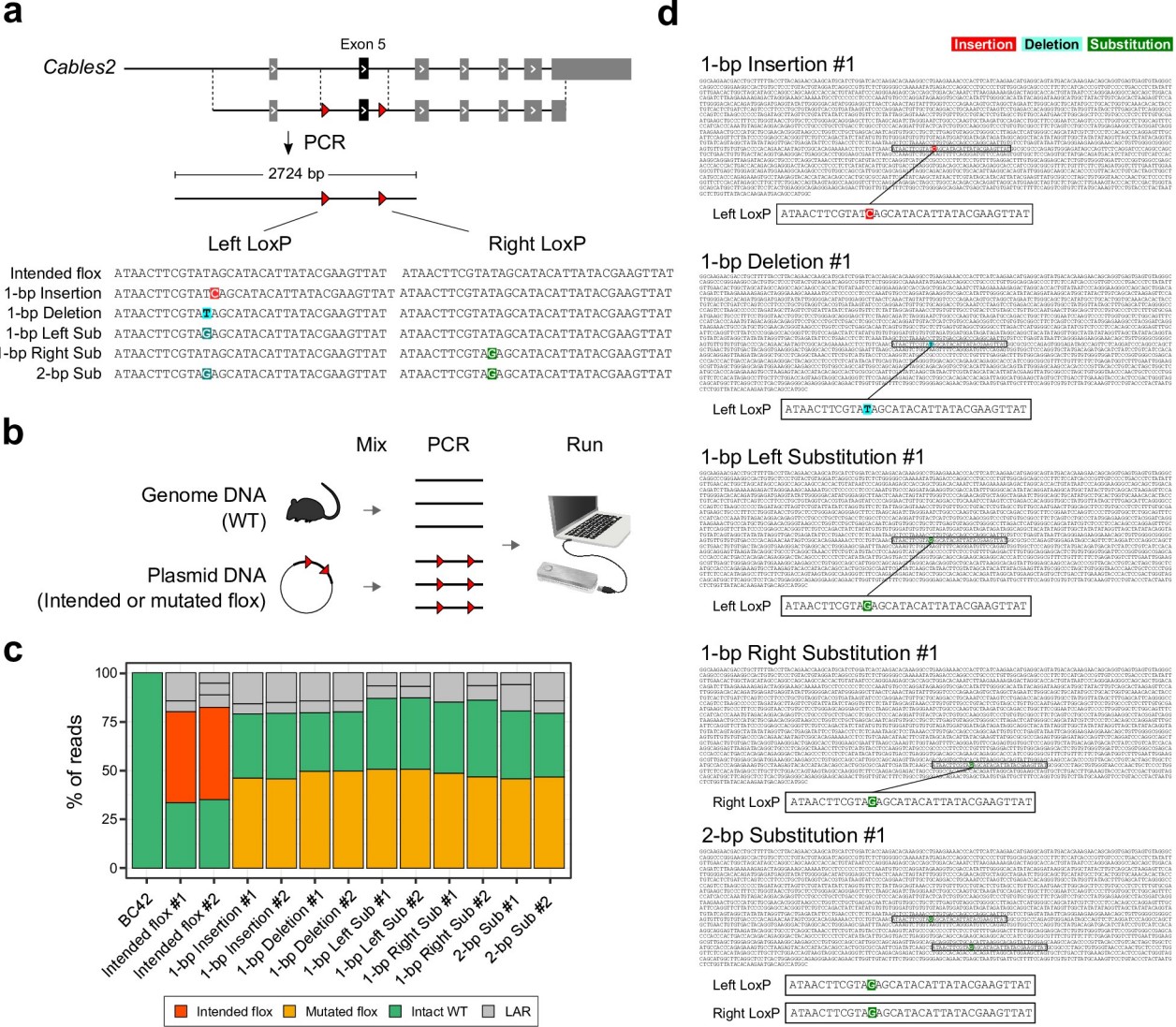

**Fig 5. Precise SNV detection in flox KI allele.** (a) Genome editing design. The red arrowheads represent LoxPs. The colours on the nucleotides represent the types of mutations, including insertion (red), deletion (sky blue), and substitution (green). (b) Experimental design. Illustrations were modified from the Togo picture gallery, licenced under CC-BY 4.0 Togo picture gallery by the DBCLS, Japan. (c) DAJIN's report of the allele percentage. The barcode numbers on the x-axis represent sample IDs. Barcode42 is a WT control. The y-axis represents the percentage of DAJIN-reported alleles. The colours of the bar represent DAJIN-reported allele types. The compartments partitioned off by horizontal lines in a bar represent the DAJIN-reported alleles. (d) DAJIN's consensus sequences of a floxed allele in each sample. The colours on the nucleotides represent mutation types, including insertion (red), deletion (sky blue), and substitution (green). The boxed sequences in the consensus sequences are LoxP sites. See S5 Data for raw data from https://osf.io/w7ade/. DAJIN, Determine Allele mutations and Judge Intended genotype by Nanopore sequencer; DBCLS, Database Center for Life Science; KI, knock-in; LAR, large rearrangement; SNV, single-nucleotide variant; Sub, substitution; WT, wild type.

DNA samples were used as a PCR template. Then, DAJIN analysed the PCR products that were 2,724 bp in length (Fig 5B). The results showed that DAJIN correctly discriminated between "WT," "Intended flox," and "Mutated flox." Furthermore, the proportion of WT and LoxP alleles reflected the designed allele frequency (Fig 5C). On the other hand, DAJIN reported there were about 20% of LAR alleles in every sample. Thus, we examined the LAR alleles and found that the LAR alleles included either the left-side LoxP or the right-side LoxP (S18 Fig). These "pseudo-LoxP" alleles could be induced by PCR recombination. Next, DAJIN's consensus sequences discovered all types of variants that we induced in the LoxPs

(Fig 5D). These results indicated that DAJIN could discriminate KI sequences according to their variants.

We next assessed DAJIN's genotyping performance using genome-edited flox KI mice. We targeted *Cables2* exon 5, and to induce it, we simultaneously cut introns 5 and 6 and knocked in the 2 LoxPs via homology-directed repair using a single-strand plasmid DNA donor. In this design, genome editing potentially generates 7 types of alleles, such as WT, flox, Left LoxP, Right LoxP, Inversion, Deletion, and Unexpected LAR alleles (Fig 6A). We obtained 20 founder mice (BC01 to BC20) and DAJIN reported that 9 mice (BC06, BC10, BC11, BC12, BC13, BC14, BC17, BC18, and BC20) contained the "Intended flox" allele, and 11 mice (BC01, BC05, BC06, BC09, BC11, BC12, BC13, BC17, BC18, BC19, and BC20) contained "Deletion" alleles (Fig 6B). Since the *Asc*I or *Eco*RV recognition sites were knocked in next to the LoxP sequence, PCR-RFLP digestion of *Asc*I or *Eco*RV can reveal LoxP insertion (Fig 6C). The PCR-RFLP results were consistent with DAJIN (Fig 6D). We also evaluated the "Deletion" alleles using standard PCR (Fig 6E), and the results were compatible with DAJIN's reports (Fig 6F). Moreover, the consensus sequence of DAJIN for BC14 showed that the entire 2,724 bp was intact, including the left and right LoxP sites (Fig 6G). Sanger sequencing also revealed that both left and right KI sequences were intact, corresponding to DAJIN's consensus sequence (Fig 6H). These results indicated that DAJIN correctly identified the intended floxed mice.

To confirm whether the next generation inherits the mutations, the BC11, BC12, BC13, and BC14, having an "Intended flox" allele were mated with WT (S19 Fig). The results showed that the genotype of the first filial generation (F1) mice from BC14 were heterozygous flox/WT, suggesting that BC14 has homozygous floxed alleles in the germline. Moreover, F1 mice from BC11, BC12, BC13, and BC18 had the "Intended flox" allele, which corresponded with DAJIN's results. Therefore, it provides evidence that DAJIN accurately captured the genotypes of the founder mice.

DAJIN detected the "Inversion" allele in 5 mice (BC02, BC05, BC07, BC10, and BC16; Fig 6B). To verify the inversion allele, we performed PCR to amplify the genomic region including the inversion junction site. The results revealed the inversion band in the same 5 samples corresponding to those mentioned in DAJIN's report (S20A and S20B Fig). Furthermore, the consensus sequence of BC02 revealed a 1-bp substitution at the inversion junction site. Sanger sequencing also detected the same substitution (S20C Fig), which suggested that DAJIN can detect complex alleles such as an inversion with SNVs.

To confirm that DAJIN is also useful for flox analysis at other loci, we further generated and analysed floxed mice for 2 genes, *Exoc7* and *Usp46*. In the *Exoc7* project (S21A Fig), we obtained 40 founder mice and analysed them using DAJIN. DAJIN identified 11 mice with the "Intended flox" allele, 7 with the "Left LoxP" allele, 16 with the "Right LoxP" allele; and 5 with the "Deletion" allele (S21B Fig). To verify DAJIN's results, we performed PCR-RFLP and standard PCR to detect the LoxP and deletion band, respectively (S22A Fig). The PCR-RFLP results agreed with the DAJIN report except for BC13 and BC32, which was shown to have no "Left LoxP" allele by DAJIN, but PCR-RFLP detected this allele (S22B Fig). Because the majority of reads in BC13 and BC32 were annotated as "Deletion" allele (S21B Fig), the deletion band might be predominantly amplified, and the number of "Left LoxP" reads decreased, owing to the PCR bias. In deletion alleles, the PCR genotyping was in agreement with DAJIN's results (S22C and S22D Fig). Next, to confirm whether the allele determined by using DAJIN was inherited through the next generation, we mated WT with *Exoc7* BC14 that DAJIN reported as heterozygous for "Intended flox" allele (42.4%) and "Right LoxP" allele (45.5%; S21B Fig). Of the total 11 F1 mice, 5 were flox/WT and 6 were Right LoxP/WT mice (S23 Fig), which represented the accuracy of DAJIN's genotyping.

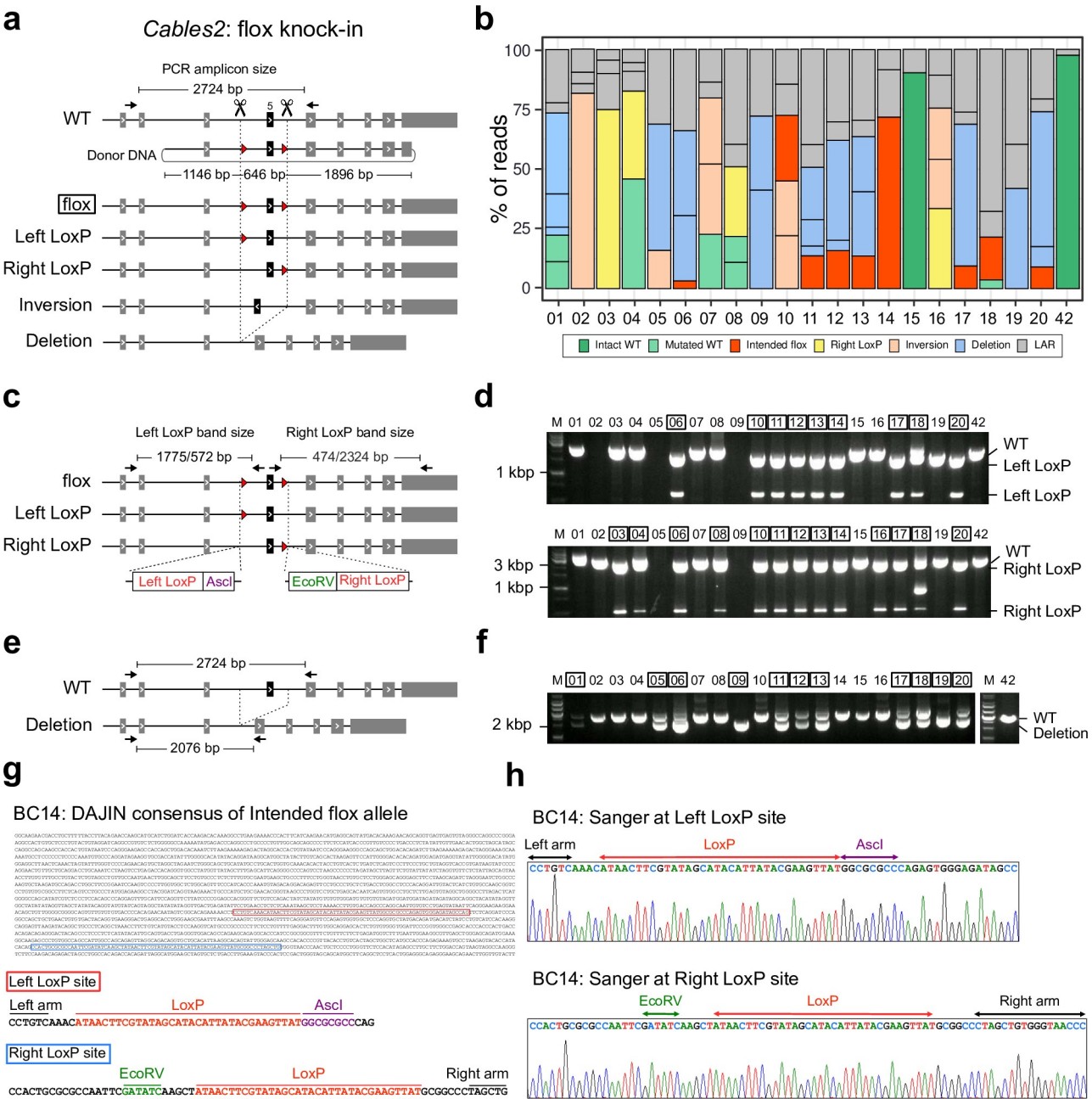

**Fig 6. Application of DAJIN for flox KI design.** (a) Genome editing design for flox KI into the Cables2 locus. The scissors represent Cas9-cutting sites. The arrows represent PCR primers, including the size of the PCR amplicon. The circular DNA represents the donor DNA. The base numbers on the donor DNA describe the left, central, and right arm sizes. The red arrowheads represent LoxPs. The boxed allele type represents the target allele. The other allele types include Left LoxP and Right LoxP. Inversion and Deletion represent possible by-products. (b) DAJIN's report of the allele percentage. The barcode numbers on the x-axis represent mouse IDs. BC42 is a WT control. The y-axis represents the percentage of DAJIN-reported alleles. The colours of the bar represent DAJIN-reported allele types. The compartments partitioned off by horizontal lines in a bar represent the DAJIN-reported alleles. (c) Design of PCR-RFLP to validate LoxP KI alleles. The AscI and EcoRV restriction sites are adjacent to Left LoxP and Right LoxP, respectively. The arrows represent PCR primers for the digested DNA fragments, including PCR product sizes. (d) PCR results for the detection of the LoxP KI allele. The top and bottom panels represent DNA fragments digested with AscI and EcoRV, respectively. The numbers on the panel mean barcode IDs. The boxed number represents the samples with LoxP alleles. (e) Design of PCR to validate deletion alleles. The arrows represent PCR primers. (f) PCR results for the detection of deletion alleles. The panels show the PCR products. The number on the panel means barcode IDs. The boxed number represents the samples with deletion alleles. (g) DAJIN's consensus sequence of the floxed allele in BC14. The red and blue boxes represent left and right LoxP sites, respectively. (h) Sanger sequences for the LoxP sites. See S6 Data for raw data from https://osf.io/w7ade/. DAJIN, Determine Allele mutations and Judge Intended genotype by Nanopore sequencer; KI, knock-in; LAR, large rearrangement; WT, wild type.

In the *Usp46* project (S24A Fig), we obtained 34 founder mice, and DAJIN reported 4 mice with the "Intended flox" allele, 2 with the "Left LoxP" allele, 2 with the "Right LoxP" allele, and 21 with the "Deletion" allele (S24B Fig). DAJIN's results were validated using PCR-RFLP that detected Left and Right LoxP alleles (S25A and S25B Fig). However, some results were inconsistent. First, PCR-RFLP analysis of BC23 and BC33 indicated that these mice may have flox alleles, but DAJIN did not report it. Second, PCR-RFLP identified "Left LoxP" alleles in BC13 and BC27, but DAJIN did not. Since DAJIN reported that these samples dominantly had the "Deletion" alleles (S24B Fig), the mismatch might be caused by PCR bias. In contrast, DAJIN detected the "Left LoxP" allele in BC21, but PCR-RLFP did not. Thus, we conducted PCR again by adjusting the dilution ratio and detected the left LoxP band (S25C Fig). For other alleles such as "Right LoxP" and "Deletion" alleles, DAJIN and PCR-RFLP's genotyping results were consistent (S25B, S25D, and S25E Fig). Notably, the PCR band in 6 samples (BC07, BC12, BC17, BC23, BC30, and BC33) seemed to be a deletion band, whereas DAJIN reported them as "LAR" alleles. Visualisation of the reads revealed that the alleles contained about 30 to 200 bp indels (S26 Fig). The result indicated that DAJIN's annotation is accurate even when distinguishing allele types by PCR band size was difficult. Finally, we investigated the next generation of mice. We obtained F1 progeny by crossing BC10 and BC11 with WT and found floxed and deletion alleles in the F1 mice (S27A Fig), which suggested that DAJIN's allele reports of BC10 and BC11 were accurate. Next, to tackle the "pseudo-LoxP" alleles, we implemented DAJIN to detect potential pseudo-LoxP alleles, and DAJIN annotated the flox allele of BC04 as a pseudo-flox allele (S27B Fig). Then, we evaluated genotype of BC04 in the F1 mice. The results revealed that the genotype of BC04 was not flox but Left-LoxP/Right-LoxP (S27B Fig), which indicates that BC04 had a pseudo-flox. Lastly, we performed NanoSV and Sniffles to test whether they can detect flox alleles. NanoSV captured the insertion, but it did not report the LoxP sequences. On the other hand, Sniffles could not detect LoxP alleles (S2 File). Taken together, these results provide evidence that DAJIN can accurately and comprehensively detect diverse mutations of floxed mice.

## Discussion

Conventional approaches such as short-range PCR, Sanger sequencing, and PCR-RFLP are standard methods to detect on-target mutagenesis induced by CRISPR-Cas and other genome editing tools. Recent studies on on-target variability of edited materials clearly show that the characterisation of genome editing events and selection of animals or cultured cells with intended and unintended mutations require alternative methods with higher sensitivity and broader range to capture mosaic mutagenic events [45,46]. In this study, we developed a genotyping method using a novel software, DAJIN, which can be applied for long-read sequencing to validate the quality of genome-edited organisms. Our method involving DAJIN has an advantage over those utilising unique molecular identifiers (UMIs) [47,48] in its automatic identification and classification of genomic rearrangements including unexpected mutations in multiple samples obtained under different editing conditions. The machine learning–based model could bypass molecular tagging to provide a feasible approach for routine assessment of genome editing outcomes.

One of DAJIN's distinguishing features is its automatic allele clustering and annotation, as well as the utilisation of a long-read sequencer. Genotyping tools similar to DAJIN have been developed previously [49–51]; however, they are optimised for short-read sequencing. Because DAJIN uses a long-read sequencer, it can identify *cis-* or *trans-*heterozygosity and complex mutant alleles such as unexpected indels and LARs. Besides, although several tools have been developed to detect PMs or LARs using long-read sequencing, DAJIN outperformed them in

capturing mutations (S10 Table, S17 Fig). It may be because the previous method focuses on monoploid or diploid organisms; however, since genome-edited samples often contain more than 2 alleles, these unpredictable allele numbers might make it challenging for the previous tools to capture mutations correctly. Polyploid phasing, which allows the reconstruction of haplotypes of the polyploid genome, is similar to DAJIN in terms of estimating alleles. WHAT-SHAP POLYPHASE [52] and H-PoPG [53] are state-of-the-art tools for polyploid phasing, but these tools require prior knowledge of the polyploidy of the target organism. Thus, the current tools for genotyping of genome-edited samples have some limitations.

Although the error rate of nanopore sequencing has improved, about 5% of errors occur in 1D sequencing of R9.4 that is the same flow cell used in our study [54]. The sequencing errors made it difficult to perform accurate allele clustering. We tackled the issue and partly solved it by (i) calculation of MIDS score (S7 Fig); (ii) reducing data's dimension by PCA; and (iii) setting proper parameters of HDBSCAN. DAJIN first converts ACGT nucleotide information to MIDS (S6 Fig). Subsequently, DAJIN subtracts the relative frequency of MIDS between a control and a sample. We called the subtracted relative frequency "MIDS score" (S7 Fig). The subtraction mitigates the sequencing errors because the error patterns are similar between a sample and a control. We next perform clustering using the MIDS score. DAJIN compresses the score by PCA and extracts 5 dimensions. The dimension reduction may be effective to mitigate sequencing errors because the sequencing errors have lower scores than true mutations. Subsequently, DAJIN performs HDBSCAN, a density-based clustering method. The HDBSCAN have a parameter of "min_cluster_size" that indicates a minimum number of samples in a cluster. DAJIN finds the parameter returning the most frequent cluster numbers by searching the value in the range of 10% to 40% of reads. It means that DAJIN ignores minor clusters that contain less than 10% of reads. We set the criteria because sequencing errors often made such minor clusters. Although the visualisation of BAM files showed reads including indel mutations in intended alleles, such as *Tyr* PM allele 2 in BC12 (Fig 2C), the Sanger sequencing revealed no mutations, which suggested that the indels visualised by the BAM file were due to nanopore sequencing errors (Fig 2D). In summary, we consider MIDS score, PCA, and the parameter setting of HDBSCAN support DAJIN's target allele detection.

As long-read sequencing induces base calling errors across a segment and cannot be used as is to validate the genome editing outcomes [22], novel screening techniques and tools need to be developed in order to identify diverse sequence changes in the genome. Short-range PCR amplification and Sanger sequencing confirmed that no additional mutation was detected in "Intact" alleles identified by DAJIN, suggesting that DAJIN validates the consequences of genome editing at the base level (Figs 2 and 4). We investigated DAJIN's accurate genotyping in 3 major genome editing designs, including PM, 2-cut KO, and flox KI, by comparing conventional methods such as PCR, RFLP, Sanger sequencing, and short-read NGS. In most cases, DAJIN's genotyping results were consistent with those obtained using the conventional methods. With regard to PM, DAJIN reported the intended PM plus LAR alleles (BC25 and BC26), whereas short-read NGS showed only the intended PM allele (Fig 3). For the 2-cut KO designs, DAJIN's genotyping results were in complete agreement with the PCR-based genotyping. In the flox design, DAJIN correctly identified intended flox KI in most cases; however, there were 3 false negative samples that were caused by PCR amplification bias (discussed in the last paragraph).

DNA double-strand break repair leads to long-range deletion, inversion, and insertion, as well as small indels in zygotes and stem cells [12,13,55–57]. In previous KI experiments, exogenous repair templates and unwanted mismatches had been identified around the target region [20,58–60]. A parallel analysis of short- and long-read sequencing results confirmed that DAJIN was able to identify editing outcomes, including unpredictable large-scale inversion

events, in mouse zygotes (S15 and S20 Figs). Short-range analysis combined with short-read NGS could not detect LAR alleles and therefore gave misleading results of founder genotype as a mosaic without LAR alleles (Fig 3B and 3C). In some cases, it recognised an LAR allele as an allele with the intended mutation (Fig 3D). In addition, it reported separate loxP insertions in *cis* generated using 2 gRNAs positioned up to 2 kb apart on the same chromosome (Figs 4 and 5, S21 and S24 Figs). Detection of mutations and/or integrations in *cis/trans* at a kilobase-scale distance requires a combination of assays and a considerable amount of time and effort. Recently, DNA cleavage in cultured cells and zygotes was shown to induce gene conversions mediated by homologous chromosomes or homologous sequences on the same chromosome [14,15]. Genotype assessment using DAJIN facilitates the selection of genome-edited samples with precisely targeted alleles or those with unwanted alleles. DAJIN might contribute to a better understanding of the consequences of editing events at the targeted locus.

Comprehensive mutation analysis might reduce the overall cost of genome editing in not only laboratory mice but also other experimental animals with a higher cost of maintenance or farm animals with a longer generation time. DAJIN is also preeminent in multispecimen processing due to its PCR-based barcoding, which enables multiplexed sequencing and allows sufficient coverage of numerous samples in a single run (S5 Table). In this study, BC01 to BC35 of Usp46 shared the same barcode as that of BC01 to BC26 of Prdm14 and BC27 to BC35 of Ddx4, and we analysed 83 samples from 3 different mouse strains in a single run. It can be undoubtedly applied to samples with a larger number of strains but a smaller number of mice for each strain. Besides, because DAJIN supports parallel processing, we were able to analyse 226 samples (total 5,982,507 reads) in only 15 h using a general-purpose desktop computer (S11 Table). Thus, DAJIN's genotyping is considerably time efficient compared to that of conventional genotyping methods. Multiple alleles may be generated in the edited cell culture pools, but they cannot be segregated as in the case of founder animals. Our results indicate that genotyping with DAJIN will add the advantage of detecting broad editing outcomes in cellular experiments (for instance, CRISPR screening) and cellular therapies where current high-throughput methods focusing on small indels may overlook longer rearrangements. Thus, DAJIN offers a novel strategy to identify multiple genomic changes, including large sequence alterations or unexpected mutations, regardless of the species or type of the material.

DAJIN's current limitations are false negatives of flox KI samples and pseudo-flox due to PCR. Although PCR enables convenient barcoding and high-level enrichment of target genomic locus, it may cause several issues. The first is PCR bias, a lower efficiency to amplify long reads and GC-rich sequences, known as length bias and GC bias. GC bias can be alleviated using high-grade DNA polymerase, but length bias cannot be removed, affecting the accuracy of DAJIN's allele percentage. In the *Exoc7* and *Usp46* flox KI, the percentage of the "Intended flox" allele was low because the deletion allele might have been preferentially amplified, and 3 false negatives (*Exoc7* BC13 and *Usp46* BC23, 33) were reported (S21 and S24 Figs). Next, the "pseudo-LoxP" alleles could be generated if the PCR products, which included one-side LoxP but not another-side LoxP, worked as a PCR primer to anneal WT allele in the next PCR step. In this study, we found that the *Usp46* BC04 included pseudo-flox alleles (S27 Fig). Since the samples with the pseudo-flox must be excluded, DAJIN flags samples that may be pseudo-flox. We still cannot exclude the possibility that LAR alleles include those generated through sequencing error and/or PCR error including these PCR-mediated recombinations. DAJIN could not identify alleles from WT mice as 100% "Intact WT" but reported that they contained a portion of "LAR" alleles in most of our experiments, which seemed to be generated artificially due to high sequencing error rate. Analysis of small portion of samples using recently developed methods may address these issues. IDMseq is used for labelling PCR amplicons using UMIs, which eliminates PCR bias and allows more quantitative analysis of allele

frequencies [61]. Karst and colleagues proposed an approach that combines dual UMI tagging with sequencing of long amplicons to generate highly accurate consensus sequences with a low PCR-mediated recombination rate [48]. nCATS enables the enrichment of the genome region without PCR; thereby, it may avoid pseudo-flox [62]. Notably, DAJIN can be used for the nanopore sequencing reads from these techniques; thus, combining these techniques with DAJIN can potentially overcome the issues caused by PCR.

## Supporting information

**S1 Fig. Simulated alleles of each genome editing design.** (a) PM design. Red box represents a target PM. Purple bar represents inserted nucleotides. (b) KO design. Black box represents a target exon. Boxed allele type represents the target allele. (c) flox KI design. Red triangles represent LoxP sequences. KI, knock-in; KO, knockout; PM, point mutation; WT, wild type.
(PDF)

**S2 Fig. Performance evaluation of abnormal allele detection.** (a) Simulated nanopore sequencing reads of abnormal alleles. The simulated read length was 2,724 bp. The integer on the exon represents the exon number. The scissor represents a Cas9-cutting site. (b) Model structure. "MIDS" and "ACGT" mean encoded reads with or without MIDS conversion, respectively. (c) UMAP visualisation of the output vectors from the FC layer. (d) The accuracy of abnormal allele detection with or without MIDS conversion. The 20 dots in each sample of x-axis represent the iteration of learning and prediction by using the DNN because the model allowed randomness. In the case of WT control, true positive means a control read is labelled as normal. The accuracy was calculated using the following formula: $accuracy = \frac{TP+TN}{TP+FP+TN+FN}$, where TP, FN, FP, and TN represent the number of true positives, false negatives, false positives, and true negatives, respectively. See S7 Data for raw data from https://osf.io/w7ade/. DNN, deep neural network; FC, fully connected; LOF, local outlier factor; MIDS, Match, Insertion, Deletion, and Substitution; UMAP, Uniform Manifold Approximation and Projection; WT, wild type; 1D CNN, one-dimensional convolutional neural network.
(PDF)

**S3 Fig. Comparison between DAJIN and SV callers.** (a) Artificial 3 alleles using simulated SV reads. (b) Comparison of DAJIN, NanoSV, and Sniffles. The alleles in bold font represent unclassified alleles. See S8 Data for raw data from https://osf.io/w7ade/. DAJIN, Determine Allele mutations and Judge Intended genotype by Nanopore sequencer; SV, structural variation.
(PDF)

**S4 Fig. Preprocessing. (**a) Screening of reads with proper sequence length and mutation loci. Grey bars represent reads. Dot bars represent deleted nucleotides. The red box represents the target mutation. The blue boxed bar represents a read exceeding the allowable length. Red dotted vertical lines represent target mutation loci. (b) MIDS conversion and one-hot encoding. The "reference" and "query" mean WT sequence and nanopore reads, respectively. MIDS, Match, Insertion, Deletion, and Substitution; WT, wild type.
(PDF)

**S5 Fig. The architecture of DNN models. (**a) Model structure. The input of the model is the encoded nanopore sequence with length (L). Three layers of a 1D-CNN include max-pooling layers and activation functions. The outputs of 1D-CNN layers are joined together into 1 vector by flattening. Each neuron in the flattened layer is attached to the FC layer. The neurons in the output layer use softmax function as the activation function, whereas all the neurons in

other layers use ReLU as the activation function. (b) Parameter setting for each layer. DNN, deep neural network; FC, fully connected; 1D CNN, one-dimensional convolutional neural network.
(PDF)

**S6 Fig. Compressed MIDS conversion.** Red and green colours represent insertion and substitution, respectively. Blue colour represents deletion. MIDS, Match, Insertion, Deletion, and Substitution.
(PDF)

**S7 Fig. MIDS score.** The black boxes represent sequences after compressed MIDS conversion. The grey boxes show representative base positions. MIDS score is calculated by subtracting the relative frequency of MIDS between a control and a sample. MIDS, Match, Insertion, Deletion, and Substitution.
(PDF)

**S8 Fig. Tyr c.140G>C, c.316G>C, and c.308G>C PM design.** The boxed nucleotides represent intended PMs. PM, point mutation; WT, wild type.
(PDF)

**S9 Fig. DAJIN's consensus sequence and Sanger sequencing of Tyr c.308G>C BC21.** The sequence represents the consensus sequence of Tyr c.308G>C BC21. The green-highlighted nucleotides represent substitution. The boxed sequences in the consensus sequences are captured by Sanger sequencing. DAJIN, Determine Allele mutations and Judge Intended genotype by Nanopore sequencer; PM, point mutation.
(PDF)

**S10 Fig. DAJIN's consensus sequence of Tyr c.230G>T PM.** The green-highlighted nucleotide represents a substitution mutation. DAJIN, Determine Allele mutations and Judge Intended genotype by Nanopore sequencer; PM, point mutation; WT, wild type.
(PDF)

**S11 Fig. PCR-based genotyping of Tyr PM design.** (a) Genome editing design. The arrows represent PCR primers for short and long PCR. (b) The short PCR results for the detection of small indel alleles. The number on the panel means barcode IDs. The asterisks represent the samples with small indels. (c) The long PCR results for the LAR detection. The number on the panel means barcode IDs. The asterisks represent the samples with LARs. LAR, large rearrangement; PM, point mutation.
(PDF)

**S12 Fig. Verification of insertion mutation of Tyr c.140G>C BC02 and BC10 mice.** (a) Visualisation of nanopore sequencing reads of BC02 and BC10. The scissor and dotted line represent a Cas-cutting site. Arrowhead represents the target nucleotide. (b) PCR design. (c) PCR results for the detection of insertion alleles. NTC, no template control; PAM, protospacer adjacent motif; WT, wild type.
(PDF)

**S13 Fig. DAJIN application to Prdm14 KO design by using Cas9 and Cas12a.** (a) Genome editing design for Prdm14 KO by using Cas9 and Cas12a. Black boxes represent exon-coding sequences numbered 6. The scissors and dotted lines represent Cas-cutting sites. The arrows represent PCR primers. The boxed allele type represents the target allele. The inversion allele represents a possible byproduct. (b) DAJIN's report of the allele percentage. The barcode numbers on the x-axis represent mouse IDs. The BC01–BC15 and BC16–BC25 are treated by

Cas12a and Cas9, respectively. The barplot of Cas9 samples is the same plot as shown in Fig 4B. BC26 is a WT control. The y-axis represents the percentage of DAJIN-reported alleles. The colours of the bar represent DAJIN-reported allele types. The horizontal lines in a bar represent the DAJIN-reported alleles. (c) Design of a short PCR to validate the target deletion allele. The arrows represent PCR primers for the digested DNA fragments, including the size of the PCR products. (d) PCR results for the detection of the target deletion allele. The number on the panel means barcode IDs. The boxed number represents the samples with deletion alleles. "Cas9" and "Cas12a" represent expected positions of deletion bands by Cas9 and Cas12a cutting. DAJIN, Determine Allele mutations and Judge Intended genotype by Nanopore sequencer; KO, knockout; LAR, large rearrangement; WT, wild type.
(PDF)

**S14 Fig. DAJIN application to Ddx4 KO design by using Cas9 and Cas12a. (**a) Genome editing design for Ddx4 KO by using Cas9 and Cas12a. The black boxes represent exon-coding sequences numbered 11–15. The scissors and dotted lines represent Cas-cutting sites. The arrows represent PCR primers. The boxed allele type represents the target allele. The inversion allele represents a possible byproduct. (b) DAJIN's report of the allele percentage. The barcode numbers on the x-axis represent mouse IDs. The BC27–BC31 and BC32–BC47 are treated by Cas12a and Cas9, respectively. BC48 is a WT control. The y-axis represents the percentage of DAJIN-reported alleles. The colours of the bar represent DAJIN-reported allele types. The horizontal lines in a bar represent the DAJIN-reported alleles. (c) PCR design to validate a target deletion allele. The arrows represent PCR primers for the digested DNA fragments, including the size of PCR products. (d) PCR results for the detection of the target deletion allele. The number on the panel means barcode IDs. The boxed number represents the samples with deletion alleles. "Cas9" and "Cas12a" represent expected positions of deletion bands by Cas9 and Cas12a cutting. DAJIN, Determine Allele mutations and Judge Intended genotype by Nanopore sequencer; KO, knockout; LAR, large rearrangement; WT, wild type.
(PDF)

**S15 Fig. DAJIN application to Stx2 KO design. (**a) Genome editing design for Stx2 KO. Shaded and black boxes represent exon-coding sequences numbered 4–6. The scissors and dotted lines represent Cas9-cutting sites. The arrows represent PCR primers. The boxed allele type represents the target allele. The inversion allele represents a possible byproduct. (b) DAJIN's report of the allele percentage. The barcode numbers on the x-axis represent mouse IDs. BC30 is a WT control. The y-axis represents the percentage of DAJIN-reported alleles. The colours of the bar represent DAJIN-reported allele types. The horizontal lines in a bar represent the DAJIN-reported alleles. (c) PCR design to validate target deletion allele. The arrows represent PCR primers for the digested DNA fragments, including the size of PCR products. (d) PCR results for the detection of the target deletion allele. The number on the panel means barcode IDs. The boxed number represents the samples with deletion alleles. (e) PCR design to validate inversion allele. The arrows represent PCR primers for the digested DNA fragments, including the size of PCR products. (f) PCR results for the detection of inversion allele. The number on the panel means barcode IDs. The boxed number represents the samples with deletion alleles. (g) Comparison between DAJIN's consensus sequence and Sanger sequencing of BC17's inversion allele. The red and purple highlighted nucleotides represent insertion and inversion, respectively. DAJIN, Determine Allele mutations and Judge Intended genotype by Nanopore sequencer; KO, knockout; LAR, large rearrangement; PAM, protospacer adjacent motif; WT, wild type.
(PDF)

**S16 Fig. Validation of DAJIN-reported LAR alleles in Stx2 BC25. (**a) PCR design to validate LAR alleles. The arrows represent PCR primers. (b) PCR results for the detection of LAR alleles. The number on the panel means barcode IDs. (c) Comparison between DAJIN's consensus sequence and Sanger sequencing. The green-highlighted nucleotide represents a substitution. DAJIN, Determine Allele mutations and Judge Intended genotype by Nanopore sequencer; LAR, large rearrangement; WT, wild type.
(PDF)

**S17 Fig. Comparison between DAJIN and SV callers.** The alleles in bold font represent misclassified alleles. See S9 Data for raw data from https://osf.io/w7ade/. DAJIN, Determine Allele mutations and Judge Intended genotype by Nanopore sequencer; SV, structural variation.
(PDF)

**S18 Fig. Pseudo-LoxP alleles.** Visualisation of simulated reads at Cables2 locus. The black boxes represent pseudo LoxPs. LAR, large rearrangement; WT, wild type.
(PDF)

**S19 Fig. Pedigree line of BC11, BC12, BC13, BC14, and BC18 in Cables2 flox KI design.** Alleles that have not been identified are marked with "*". KI, knock-in; LAR, large rearrangement; WT, wild type.
(PDF)

**S20 Fig. Validation of DAJIN-reported inversion alleles. (**a) PCR design to validate inversion alleles. The arrows represent PCR primers for the digested DNA fragments, including the size of PCR products. (b) PCR results for the detection of inversion alleles. The number on the panel means barcode IDs. The boxed number represents the samples with inversion alleles. (c) Comparison between DAJIN's consensus sequence and Sanger sequencing. The sequence represents the consensus sequence of inversion alleles of BC02. The green-highlighted nucleotides represent substitution. The dotted lines represent corresponding nucleotides between Sanger and DAJIN's consensus sequences. DAJIN, Determine Allele mutations and Judge Intended genotype by Nanopore sequencer; WT, wild type.
(PDF)

**S21 Fig. DAJIN application to Exoc7 flox KI design. (**a) Genome editing design for flox KI into the Exoc7 locus. The scissors represent Cas9-cutting sites. The arrows represent PCR primers. The circular DNA represents the donor DNA. The base numbers on the donor DNA describe the size of the left, central, and right arms. The red arrowheads represent LoxPs. The boxed allele type represents the target allele. The other allele types include Left LoxP and Right LoxP. Inversion and Deletion represent possible byproducts. (b) DAJIN's report of the allele percentage. The barcode numbers on the x-axis represent mouse IDs. The BC41 is a WT control. The y-axis represents the percentage of DAJIN-reported alleles. The colours of the bar represent DAJIN-reported allele types. The horizontal lines in a bar represent the DAJIN-reported alleles. DAJIN, Determine Allele mutations and Judge Intended genotype by Nanopore sequencer; KI, knock-in; LAR, large rearrangement; WT, wild type.
(PDF)

**S22 Fig. PCR-based genotyping of Exoc7 flox KI design. (**a) PCR-RFLP design to validate LoxP KI alleles. The AscI and EcoRV digest the restriction sites adjacent to Left LoxP and Right LoxP, respectively. The arrows represent PCR primers for the digested DNA fragments, including PCR product sizes. (b) PCR results for the detection of LoxP KI alleles. The top and bottom panels represent the DNA fragments digested with AscI and EcoRV, respectively. The number on the panel means barcode IDs. The boxed number represents the samples with

LoxP alleles. The asterisks represent mismatched samples from DAJIN's genotyping. (c) PCR design to validate deletion alleles. The arrows represent PCR primers for the digested DNA fragments, including the size of PCR products. (d) PCR results for the detection of deletion alleles. The number on the panel means barcode IDs. The boxed number represents the samples with deletion alleles. DAJIN, Determine Allele mutations and Judge Intended genotype by Nanopore sequencer; KI, knock-in; WT, wild type.
(PDF)

**S23 Fig. Pedigree line of BC14 in Exoc7 flox KI design.** KI, knock-in; WT, wild type.
(PDF)

**S24 Fig. DAJIN application to Usp46 flox KI design. (**a) Genome editing design for flox KI into the Usp46 locus. The scissors represent Cas9-cutting sites. The arrows represent PCR primers including the size of PCR amplicon. The circular DNA represents the donor DNA. The base numbers on the donor DNA describe the size of the left, central, and right arms. The red arrowheads represent LoxPs. The boxed allele type represents the target alleles. The other allele types include Left LoxP and Right LoxP. Inversion and Deletion represent possible byproducts. (b) DAJIN's report of the allele percentage. The barcode numbers on the x-axis represent mouse IDs. The BC35 is a WT control. The y-axis represents the percentage of DAJIN-reported alleles. The colours of the bar represent DAJIN-reported allele types. The horizontal lines in a bar represent the DAJIN-reported alleles. The asterisk on BC04 represents a pseudo-flox mouse. DAJIN, Determine Allele mutations and Judge Intended genotype by Nanopore sequencer; KI, knock-in; LAR, large rearrangement; WT, wild type.
(PDF)

**S25 Fig. PCR-based genotyping of Usp46 flox KI design. (**a) PCR-RFLP design to validate LoxP KI alleles. The AscI and EcoRV digest the restriction sites adjacent to Left LoxP and Right LoxP, respectively. The arrows represent PCR primers for the digested DNA fragments, including PCR product sizes. (b) PCR results for the detection of LoxP KI alleles. The top and bottom panels represent the DNA fragments digested with AscI and EcoRV, respectively. The number on the panel means barcode IDs. The boxed number represents the samples with LoxP alleles. "M" and "B" means marker and blank, respectively. (c) PCR results for the detection of DAJIN-reported left LoxP alleles in BC21. The number on the panel means barcode IDs and its dilution condition. The boxed number represents the samples with left LoxP alleles. (d) PCR design to validate deletion alleles. The arrows represent PCR primers for the digested DNA fragments, including the size of PCR products. (e) PCR results for the detection of deletion alleles. The number on the panel means barcode IDs. The boxed number represents the samples with deletion alleles. DAJIN, Determine Allele mutations and Judge Intended genotype by Nanopore sequencer; KI, knock-in; WT, wild type.
(PDF)

**S26 Fig. DAJIN distinguishes between deletion and LAR alleles.** DAJIN labelled the BC07, BC12, BC17, BC23, BC30, and BC33 as "LAR." The BC06 is a "Deletion" allele as a control. DAJIN, Determine Allele mutations and Judge Intended genotype by Nanopore sequencer; LAR, large rearrangement.
(PDF)

**S27 Fig. Pedigree line of BC10 and BC11 in Usp46 flox KI design. (**a) The flox mouse line, which transmitted to its pedigree. (b) The pseudo-flox mouse line. Alleles that have not been identified are marked with "*". KI, knock-in; WT, wild type.
(PDF)

**S1 Table. gRNA and ssODN sequence.**
(XLSX)

**S2 Table. Mouse production process.**
(XLSX)

**S3 Table. Target amplicon primers for nanopore sequencing.**
(XLSX)

**S4 Table. Barcode attachment primers for nanopore sequencing.**
(XLSX)

**S5 Table. Read number of each sample.**
(XLSX)

**S6 Table. Genotyping PCR primer.**
(XLSX)

**S7 Table. PCR target amplicon primer for next-generation sequencing.**
(XLSX)

**S8 Table. Model performance.**
(XLSX)

**S9 Table. Comparison DAJIN to NGS and SNV callers.**
(XLSX)

**S10 Table. Comparison DAJIN to NGS and SNV callers.**
(XLSX)

**S11 Table. Processing time and computational resources.**
(XLSX)

**S1 File. DAJIN's consensus sequences (HTML).**
(ZIP)

**S2 File. VCF files by NanoSV and Sniffles for Prdm14 and Cables2.**
(ZIP)

**S1 Raw images. Uncropped gel images from all main and Supporting information figures.**
(PDF)

## Acknowledgments

We would like to thank the staff at the Laboratory Animal Resource Center University of Tsukuba for their help in breeding and rearing of the mice. We are grateful to Tomoyuki Fujiyama for advice on the experimental design. We would also like to thank Ozaki Haruka for fruitful discussions.

## Author Contributions

**Conceptualization:** Akihiro Kuno, Shinya Ayabe, Kazuya Murata, Fumihiro Sugiyama, Seiya Mizuno.

**Data curation:** Sayaka R. Suzuki.

**Funding acquisition:** Akihiro Kuno, Atsushi Yoshiki, Satoru Takahashi, Seiya Mizuno.

**Investigation:** Akihiro Kuno, Yoshihisa Ikeda, Kanako Kato, Kento Morimoto, Arata Wakimoto, Natsuki Mikami, Megumi Takemura, Tra Thi Huong Dinh, Masafumi Muratani.

**Methodology:** Akihiro Kuno, Yoko Tanimoto.

**Project administration:** Shinya Ayabe, Atsushi Yoshiki, Fumihiro Sugiyama, Satoru Takahashi, Seiya Mizuno.

**Resources:** Natsuki Mikami, Miyuki Ishida, Natsumi Iki, Yuko Hamada, Megumi Takemura, Yoko Daitoku, Yoko Tanimoto, Michito Hamada, Seiya Mizuno.

**Software:** Akihiro Kuno, Kotaro Sakamoto.

**Supervision:** Seiya Mizuno.

**Validation:** Kazuya Murata.

**Writing – original draft:** Akihiro Kuno, Yoshihisa Ikeda, Shinya Ayabe, Kotaro Sakamoto, Sayaka R. Suzuki, Seiya Mizuno.

**Writing – review & editing:** Fumihiro Sugiyama, Satoru Takahashi.

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
