## [Editor Report · Decision Letter 0]

4 Nov 2021

Dear Dr Mizuno, 

Thank you for submitting your manuscript entitled "Multiplex genotyping method to validate the multiallelic genome editing outcomes using machine learning-assisted long-read sequencing" for consideration as a Methods and Resources article by PLOS Biology. Please accept our apologies for the delay in sending the decision below to you.

Your manuscript has now been evaluated by the PLOS Biology editorial staff, as well as by an academic editor with relevant expertise, and I am writing to let you know that we are positive about the way you have addressed the critiques raised by the referees at Review Commons.

Therefore, we would like to invite you to complete your submission by providing the metadata that is required for full assessment. We will then decide if we need to seek further advice from the reviewers.

To complete your submission, please login to Editorial Manager where you will find the paper in the 'Submissions Needing Revisions' folder on your homepage. Please click 'Revise Submission' from the Action Links and complete all additional questions in the submission questionnaire.

Once your full submission is complete, your paper will undergo a series of checks in preparation for peer review. Once your manuscript has passed the checks, we might send it out for review. 

Please re-submit your manuscript within two working days, i.e. by Nov 08 2021 11:59PM.

Kind regards,

Gabriel Gasque, Senior Editor

on behalf of

Richard Hodge

Associate Editor

PLOS Biology

rhodge@plos.org

---

## [Editor Report · Decision Letter 1]

19 Nov 2021

Dear Dr Mizuno,

Thank you for submitting your manuscript "Multiplex genotyping method to validate the multiallelic genome editing outcomes using machine learning-assisted long-read sequencing" for consideration as a Methods and Resources article by PLOS Biology. Please accept my apologies for the slight delay in getting back to you as the academic editor looked over the revision of your manuscript and the responses to the reviewer comments from Review Commons.

We have now discussed the manuscript with the academic editor, who is satisfied that the reviewer concerns have been fully addressed. Based on this, we will probably accept this manuscript for publication, provided you address the following data and other policy-related requests that I have provided below:

A) We would to suggest the following modification to the title, to make it more compelling and accessible for our broad readership:

‘DAJIN enables multiplex genotyping to simultaneously validate on- and off-target genome editing outcomes’

B) In your ethics statement in the Methods section of the manuscript, please include the full name of the Institutional Animal Care and Use Committee (IACUC) that reviewed and approved the animal care and use protocol/permit/project license. Alternatively, please confirm that ‘Institutional Animal Experiment Committee of the University of Tsukuba’ is an IACUC. Please also include the IACUC approval number and state the method of euthanasia used in the animal studies. 

C) You may be aware of the PLOS Data Policy, which requires that all data be made available without restriction: http://journals.plos.org/plosbiology/s/data-availability. For more information, please also see this editorial: http://dx.doi.org/10.1371/journal.pbio.1001797

- Supplementary files (e.g., excel). Please ensure that all data files are uploaded as 'Supporting Information' and are invariably referred to (in the manuscript, figure legends, and the Description field when uploading your files) using the following format verbatim: S1 Data, S2 Data, etc. Multiple panels of a single or even several figures can be included as multiple sheets in one excel file that is saved using exactly the following convention: S1_Data.xlsx (using an underscore).

- Deposition in a publicly available repository. Please also provide the accession code or a reviewer link so that we may view your data before publication.

Regardless of the method selected, please ensure that you provide the individual numerical values that underlie the summary data for the following Figures, as they are essential for readers to assess your analysis and to reproduce it:

Figure 1B, 2B-C, 2E, 2G, 3A-E, 4B-C, 4E, 5C, 6B, 6H, S2C-D, S3, S17

D) Please also ensure that each of the relevant figure legends in your manuscript include information on *WHERE THE UNDERLYING DATA CAN BE FOUND*, and ensure your supplemental data file/s has a legend

E) We require the original, uncropped and minimally adjusted images supporting all blot and gel results reported in the following Figures:

Figure 6D, 6F, S11B-C, S12C, S13C, S14D, S15D, S15F, S16B, S20B, S22B, S22D, S25B-C, S25E

We will require these files before a manuscript can be accepted so please prepare and upload them now. Please carefully read our guidelines for how to prepare and upload this data: https://journals.plos.org/plosbiology/s/figures#loc-blot-and-gel-reporting-requirements.

We expect to receive your revised manuscript within two weeks.

*Published Peer Review History*

*Early Version*

Sincerely,

Richard

Richard Hodge, PhD

Associate Editor, PLOS Biology

rhodge@plos.org

---

## [Editor Report · Decision Letter 2]

7 Dec 2021

Dear Dr Mizuno,

On behalf of my colleagues and the Academic Editor, Bon-Kyoung Koo, I am pleased to say that we can in principle accept your Methods and Resources article "DAJIN enables multiplex genotyping to simultaneously validate intended and unintended target genome editing outcomes" for publication in PLOS Biology, provided you address any remaining formatting and reporting issues. These will be detailed in an email that will follow this letter and that you will usually receive within 2-3 business days, during which time no action is required from you. Please note that we will not be able to formally accept your manuscript and schedule it for publication until you have any requested changes.

PRESS

Sincerely, 

Richard

Richard Hodge, PhD

Associate Editor, PLOS Biology

rhodge@plos.org

PLOS
